# Joint Embedding Variational Bayes

**Amin Oji**                                                                                       *amin.oji@uwaterloo.ca*
*Department of Systems Design Engineering*
*University of Waterloo*
*Waterloo, Canada*

**Paul Fieguth**                                                                                 *paul.fieguth@uwaterloo.ca*
*Department of Systems Design Engineering*
*University of Waterloo*
*Waterloo, Canada*

**Reviewed on OpenReview:** *https://openreview.net/forum?id=4cbPJ5jLtr*

## Abstract

We introduce Variational Joint Embedding (VJE), a reconstruction-free latent-variable framework for non-contrastive self-supervised learning in representation space. VJE maximizes a symmetric conditional evidence lower bound (ELBO) on paired encoder embeddings by defining a conditional likelihood directly on target representations, rather than optimizing a pointwise compatibility objective. The likelihood is instantiated as a heavy-tailed Student–$t$ distribution on a polar representation of the target embedding, where a directional–radial decomposition separates angular agreement from magnitude consistency and mitigates norm-induced pathologies. The directional factor operates on the unit sphere, yielding a valid variational bound for the associated spherical subdensity model. An amortized inference network parameterizes a diagonal Gaussian posterior whose feature-wise variances are shared with the directional likelihood, yielding anisotropic uncertainty without auxiliary projection heads. Across ImageNet–1K, CIFAR–10/100, and STL-10, VJE is competitive with standard non-contrastive baselines under linear and $k$-NN evaluation, while providing probabilistic semantics directly in representation space for downstream uncertainty-aware applications. We validate these semantics through out-of-distribution detection, where representation-space likelihoods yield strong empirical performance. These results position the framework as a principled variational formulation of non-contrastive learning, in which structured feature-wise uncertainty is represented directly in the learned embedding space.

## 1 Introduction

Joint embedding architectures have emerged as a powerful paradigm for self-supervised representation learning in computer vision. These architectures can be broadly categorized into contrastive and non-contrastive methods. Contrastive approaches, such as SimCLR (Chen et al., 2020a) and MoCo (He et al., 2020; Chen et al., 2020b), learn representations by maximizing similarity between pairs of semantically related (positive) samples, and minimizing similarity between unrelated (negative) samples. Since these methods are unsupervised, they must approximate 'negative' associations through heuristic choices of negatives (typically other instances in the minibatch, and in some cases external memory banks or queues) to ensure the presence of sufficiently diverse samples. This estimation procedure increases computational and memory demands, and can complicate training in domains where modelling patterns can be sensitive to erroneous negative associations.

Non-contrastive methods, including BYOL (Grill et al., 2020), SimSiam (Chen and He, 2021), VICReg (Bardes et al., 2022), and Barlow Twins (Zbontar et al., 2021), avoid the need for negative samples by learning from paired views of the same input. To prevent the model from collapsing to a trivial solution (e.g., mapping all inputs to the same constant representation), these approaches rely on architectural asymmetries,

auxiliary heads (e.g., prediction/projection heads), or redundancy-reduction objectives. By eliminating the dependence on negative samples, non-contrastive methods simplify training and can extend the applicability of self-supervised learning to settings where the notion of similarity and what constitutes a 'negative' sample are inherently ambiguous. A prominent viewpoint within this landscape is the Joint Embedding Predictive Architecture (JEPA) (LeCun, 2022), which formulates training in terms of minimizing a pointwise compatibility or energy between predicted and target embeddings in representation space, with recent instantiations including I-JEPA (Assran et al., 2023) and LeJEPA (Balestriero and LeCun, 2025).

Despite their empirical success, both contrastive and non-contrastive objectives are commonly trained to produce deterministic point embeddings, as each input is mapped to a single point in latent space and learning proceeds by optimizing pointwise similarity or discrepancy objectives between paired embeddings. As a result, such representations are not typically optimized as tractable probabilistic models in representation space, and therefore do not readily provide likelihood-based scoring or feature-wise uncertainty. In applications such as medical diagnosis (Begoli et al., 2019; Ghesu et al., 2019; Gal et al., 2017), anomaly and out-of-distribution detection (Schlegl et al., 2019; Zimmerer et al., 2018; Wang and Huang, 2018; Liang et al., 2018), and reinforcement learning (Depeweg et al., 2018; Chua et al., 2018; Ha and Schmidhuber, 2018), the inability to represent uncertainty over latent factors can limit the reliability, interpretability, and downstream utility of the learned representations.

Variational methods, and in particular Variational Autoencoders (VAEs) (Kingma and Welling, 2014) and their extensions (Higgins et al., 2017; van den Oord et al., 2017; Maaløe et al., 2019), provide a principled probabilistic framework by modelling latent variables as distributions. This is accomplished through a reconstruction-based objective, which ensures that latent variables capture detailed generative factors associated with pixel-level input data. However, when the end goal is to obtain high-level semantic representations for downstream tasks, enforcing pixel-level fidelity can impose significant and often unnecessary computational overhead. This motivates latent-variable objectives defined directly on representations rather than on pixels, while retaining the probabilistic structure that makes variational methods attractive in uncertainty-sensitive applications.

In this work, we introduce **Variational Joint Embedding (VJE)**, a framework that synthesizes **variational inference** and **joint embedding** to provide a latent-variable formulation of non-contrastive self-supervised learning without relying on input reconstruction (i.e., pixel-level) or contrastive objectives. In contrast to pointwise compatibility objectives that directly optimize discrepancies between embeddings, VJE defines an explicit conditional likelihood on target embeddings and trains by maximizing a symmetric conditional evidence lower bound (ELBO). The likelihood is modelled with a heavy-tailed Student–$t$ distribution on the reparameterized target observation $y = (\hat{\mathbf{z}}, \|\mathbf{z}\|)$, where a directional–radial decomposition separates angular agreement from magnitude consistency and mitigates norm-induced pathologies. The directional component is motivated by a normalized spherical construction and realized in practice on the unit sphere, yielding a subnormalized directional term that defines a valid variational lower bound for the associated subprobability model. An amortized inference network parameterizes a variational posterior over the corresponding latent variables, with feature-wise variances shared between the posterior and the directional term. The probabilistic semantics developed here operate at the level of the representation space itself, making likelihood-based scoring and feature-wise uncertainty direct properties of the learned representations rather than post-hoc additions. The target branch is treated as fixed within each update, implemented by stop-gradient or equivalently by a target encoder held fixed during the update, including EMA, to implement conditional likelihood semantics. We formalize this objective-level distinction between pointwise energy-based predictive losses and likelihood-based training in Appendix B.

Empirically, VJE is competitive with strong non-contrastive baselines on representation learning benchmarks across ImageNet (Russakovsky et al., 2015), STL-10 (Coates et al., 2011), and CIFAR (Krizhevsky, 2009), while learning probabilistic representations with likelihood and uncertainty semantics in representation space. We validate these semantics through out-of-distribution detection, where representation-space likelihoods yield strong empirical performance. Taken together, these results position VJE as a principled probabilistic formulation of non-contrastive learning, with likelihood-based modelling as the underlying training primitive and feature-wise uncertainty represented directly in latent space.

## 2 Background

**Non-contrastive self-supervised learning (SSL).** The central principle of non-contrastive SSL is to relate representations derived from different views of the same input, without relying on negative samples. Under the JEPA viewpoint (LeCun, 2022; Assran et al., 2023), this is instantiated as prediction in representation space under a pointwise compatibility objective. More specifically, a context representation is used to predict a corresponding target representation from a paired view. Formally, given two related views $x_1$ and $x_2$, the encoder $f_\theta$ and predictor $g_\phi$ are learned by minimizing:

$$\mathcal{L}_{\text{JEPA}}(x_1, x_2) = d\big(g_\phi(f_\theta(x_1)),\, f_\xi(x_2)\big), \tag{1}$$

where $f_\xi$ denotes a target encoder providing target embeddings for the predictive loss, and the metric $d(\cdot, \cdot)$ is typically a distance or similarity measure (e.g., cosine or Euclidean distance). Many non-contrastive objectives can be written in closely related paired-view forms, including BYOL (Grill et al., 2020) and SimSiam (Chen and He, 2021), which rely on asymmetric branches, as well as VICReg (Bardes et al., 2022) and Barlow Twins (Zbontar et al., 2021), which impose explicit variance/covariance or redundancy-reduction penalties. Despite their empirical success, these approaches produce deterministic embeddings by mapping each input to a single point in latent space, and their pointwise energy or compatibility primitives are not themselves given by a tractable normalized probabilistic model in representation space (LeCun, 2022).

**Variational inference and uncertainty quantification.** Variational inference provides a general framework for probabilistic representation learning by representing each input as a distribution over latent factors. A canonical instance is the Variational Autoencoder (VAE) (Kingma and Welling, 2014), which trains an encoder to produce an approximate posterior over latent variables and a decoder to reconstruct the input, jointly optimized by maximizing an evidence lower bound (ELBO) that balances a reconstruction likelihood against KL regularization toward a prior (Kingma and Welling, 2014; Rezende et al., 2014). The reparameterization trick enables gradient-based optimization by expressing stochastic samples as differentiable transformations of noise. While this establishes a rigorous and tractable probabilistic foundation, its dependence on pixel-level reconstruction motivates alternative approaches suited to representation learning settings where such reconstruction is not required.

Beyond reconstruction-based models, variational and uncertainty-aware objectives have been explored in contrastive, supervised, and deterministic settings. Yavuz and Yanikoglu (2024) propose a variational contrastive objective using beta-divergence, Wang et al. (2025) develop variational supervised contrastive learning with label-conditioned priors, Jeong et al. (2025) introduce probabilistic variational contrastive learning, and SNGP (Liu et al., 2020) provides uncertainty estimates through distance-aware feature spaces with a Gaussian-process output layer. These approaches share the motivation of integrating uncertainty into representation learning, but operate under different supervision regimes or objective classes than the non-contrastive setting we target.

**Variational inference in joint embedding architectures.** Recent works have attempted to combine the efficiency of non-contrastive self-supervised learning with the probabilistic foundations of variational inference to produce uncertainty-aware representations, though several challenges remain. Notably, VI-SimSiam (Nakamura et al., 2023) modifies SimSiam by wrapping each unit-length embedding in a Power-Spherical (PS) (Cao et al., 2020) density parameterized by a mean direction $\boldsymbol{\mu}_i$ and concentration $\kappa_i$, with one branch's embedding frozen via stop-gradient and $\kappa_i = u_\theta(x_i)$ predicted by an additional scalar head. The resulting loss is defined as a PS log-likelihood applied to unit-norm embeddings:

$$\mathcal{L}_{\text{align}} = \tfrac{1}{2}\Big[ -\log \text{PS}\big(\boldsymbol{z}_2; \boldsymbol{\mu}_1, \kappa_1\big) - \log \text{PS}\big(\boldsymbol{z}_1; \boldsymbol{\mu}_2, \kappa_2\big)\Big]. \tag{2}$$

In their variational interpretation, this likelihood term is augmented with a KL divergence between the PS density and a hyperspherical prior on $\mathbb{S}^{D-1}$, yielding an ELBO-like objective on unit embeddings. Mathematically, this defines a directional density on the unit sphere, with the scalar concentration $\kappa_i$ controlling dispersion around $\boldsymbol{\mu}_i$. This is a coherent probabilistic formulation of non-contrastive alignment, but its uncertainty mechanism is inherently limited because $\kappa_i$ is a single scalar. Decreasing $\kappa_i$ pushes the

density toward the uniform distribution on $\mathbb{S}^{D-1}$, directly weakening the alignment penalty on hard or noisy pairs. As a result, the learned $\kappa_i$ can largely function as an example-dependent temperature controlling how strongly directional agreement is enforced, rather than as a structured uncertainty representation. Moreover, the scalar form restricts uncertainty to isotropic dispersion on the sphere and cannot represent more structured, direction-dependent uncertainty.

VSSL (Yavuz and Yanikoglu, 2025) takes a different approach, coupling a student encoder with a momentum-updated teacher, both outputting diagonal Gaussians over latent features. The teacher processes view $x_1$ to define a data-dependent "prior" $p_{\theta_t}(\boldsymbol{s} \mid x_1) = \mathcal{N}(\boldsymbol{s}; \boldsymbol{\mu}_1, \boldsymbol{\sigma}_1^2)$, while the student processes view $x_2$ to produce a "posterior" $q_{\theta_s}(\boldsymbol{s} \mid x_2) = \mathcal{N}(\boldsymbol{s}; \boldsymbol{\mu}_2, \boldsymbol{\sigma}_2^2)$. The objective combines a likelihood term evaluating student samples under the teacher Gaussian:

$$\mathbb{E}_{q(\boldsymbol{s}|x_2)}\big[\log \mathcal{N}\big(\boldsymbol{s}; \boldsymbol{\mu}_1, \boldsymbol{\sigma}_1^2\big)\big], \tag{3}$$

with a KL penalty $\mathrm{KL}\big(q_{\theta_s}(\boldsymbol{s} \mid x_2) \,\|\, p_{\theta_t}(\boldsymbol{s} \mid x_1)\big)$. This is framed as a self-supervised ELBO with the teacher as prior. However, because the teacher is defined by a moving-average update of the student, the KL term primarily enforces teacher–student consistency rather than regularization toward a fixed Bayesian prior.

More fundamentally, replacing the analytic Gaussian KL and log-likelihood with cosine-based alternatives weakens the probabilistic interpretation: these cosine quantities depend only on the directions of the mean and variance vectors and are not the KL divergence or log-likelihood of a Gaussian in $\mathbb{R}^D$. The resulting objective is therefore better viewed as a heuristic angular-alignment loss, insensitive to parameter norms and misaligned with the Euclidean geometry underlying the Gaussian distribution. Consequently, the variance parameters are optimized as directional features rather than calibrated uncertainty measures, and their probabilistic semantics become unclear.

Collectively, these efforts highlight the potential of integrating variational inference into non-contrastive joint embedding architectures, while also demonstrating the difficulty of obtaining coherent probabilistic semantics when pointwise compatibility objectives remain the underlying training primitive. This motivates our work, in which the latent-variable model is constructed directly in representation space from the outset.

## 3 Model Architecture

Our implementation of the Variational Joint Embedding (VJE) framework follows the standard training structure of non-contrastive self-supervised learning (Chen and He, 2021; Grill et al., 2020), while adopting a probabilistic latent-variable objective. A stochastic augmentation $\tau$ is applied twice to an input $x$ to produce two views, $x_1 = \tau^{(1)}(x)$ and $x_2 = \tau^{(2)}(x)$, which are processed by a shared encoder $f_\theta : \mathcal{X} \to \mathbb{R}^D$ into deterministic embeddings $\mathbf{z}_1 = f_\theta(x_1)$ and $\mathbf{z}_2 = f_\theta(x_2)$. The framework consists of two asymmetric branches: an *inference branch* that maps $\mathbf{z}_i$ to the parameters of a stochastic latent code for view $i$, and a *target branch* that treats the opposite embedding $\mathbf{z}_j$ as a fixed observation during training. This fixed-observation semantics can be implemented by stop-gradient or equivalently by a target encoder held fixed during each update, including exponential moving average (EMA) variants. This asymmetry is used for stable training, as in other non-contrastive methods, but in VJE it additionally carries a precise semantic role as the fixed-observation conditioning required by the conditional ELBO derived in Section 4.

Each embedding is represented by its unit direction $\hat{\mathbf{z}}_i = \mathbf{z}_i/\|\mathbf{z}_i\|$ and magnitude $\|\mathbf{z}_i\|$, yielding the representation-space observation $y_i = (\hat{\mathbf{z}}_i, \|\mathbf{z}_i\|)$. The inference network produces a variational posterior $q_i(\mathbf{s}) \equiv q_\phi(\mathbf{s} \mid \mathbf{z}_i)$, from which a latent sample $\mathbf{s}_i$ is drawn and evaluated against the target observation $y_j$. The radial residual $\Delta r_{ij} = \|\mathbf{z}_j\| - \|\mathbf{s}_i\|$ measures the magnitude discrepancy between the target embedding and the latent sample. This separation allows directional and norm discrepancies to be modelled with dedicated likelihood terms adapted to the geometry of the embedding space. We factorize the conditional likelihood into a directional term and a one-dimensional radial term, both instantiated as heavy-tailed Student–$t$ likelihoods as detailed in Section 4. The overall architecture is depicted in Figure 1.

**Encoder and inference network.** The encoder $f_\theta : \mathcal{X} \to \mathbb{R}^D$ is a shared backbone across views, and an inference network $g_\phi$ maps each $\mathbf{z}_i$ to the parameters of a diagonal Gaussian variational posterior, $q_i(\mathbf{s}) = \mathcal{N}\big(\boldsymbol{\mu}_i, \ \mathrm{diag}(\boldsymbol{\sigma}_i^2)\big)$, with $q_i(\mathbf{s}) \equiv q_\phi(\mathbf{s} \mid \mathbf{z}_i)$. While $g_\phi$ is architecturally akin to the predictor networks

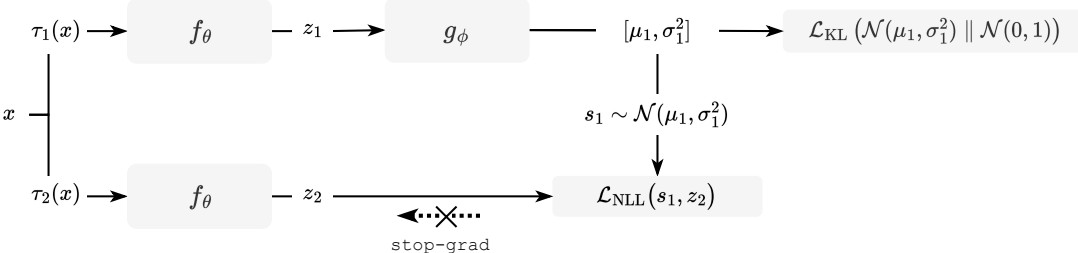

Figure 1: The asymmetric forward pass for one conditional direction in VJE, from view 1 to view 2. An encoder $f_\theta$ produces $\mathbf{z}_1$, and an amortized inference network $g_\phi$ maps it to a latent distribution $q_1(\mathbf{s}) = \mathcal{N}(\boldsymbol{\mu}_1, \boldsymbol{\sigma}_1^2)$. A sample $\mathbf{s}_1$ is drawn and the conditional likelihood of the target observation $y_2 = (\hat{\mathbf{z}}_2, \|\mathbf{z}_2\|)$ is evaluated under this latent code, with the radial residual $\Delta r_{12} = \|\mathbf{z}_2\| - \|\mathbf{s}_1\|$ scoring the magnitude discrepancy. The target branch is detached, enforcing fixed-observation semantics for the conditional likelihood term. The loss consists of directional ($\ell_{\mathrm{dir}}$) and radial ($\ell_{\mathrm{rad}}$) negative log-likelihoods (NLLs), jointly denoted $\mathcal{L}_{\mathrm{NLL}}$, together with a Kullback–Leibler (KL) divergence term $\mathcal{L}_{\mathrm{KL}}$.

used in non-contrastive methods, we refer to it as an *amortized inference network* to reflect its variational role, since it parameterizes an instance-conditional posterior from which reparameterized samples are drawn and evaluated under the conditional likelihood. The network $g_\phi$ is implemented as a bottleneck MLP in which each layer applies a linear transformation followed by layer normalization and a nonlinear activation function. Its final hidden representation is mapped through two linear output heads to produce $\boldsymbol{\mu}_i$ and $\boldsymbol{\sigma}_i^2$.

The same variance vector $\boldsymbol{\sigma}_i^2$ is used both as the diagonal covariance of the variational posterior $q_i$ and, via $\Sigma = \mathrm{diag}(\boldsymbol{\sigma}_i^2)$, as the scale matrix in the directional Student–$t$ term. This tying ensures that feature-wise dispersion governs both the posterior sampling distribution and the directional likelihood in a consistent manner. In high-dimensional settings, a centered version of this variance may additionally be used within the directional term for numerical stability, as discussed in Appendix A.2. We adopt a diagonal covariance to keep the number of parameters linear in $D$ and to ensure that sampling and KL evaluation scale linearly with the embedding dimension.

Unlike other joint-embedding architectures (Chen and He, 2021; Bardes et al., 2022; Grill et al., 2020; Assran et al., 2023), VJE does not introduce a separate projection head, as doing so would define an auxiliary representation space whose geometry is not constrained to relate to that of the encoder output. Instead, both the conditional likelihood and the variational posterior are defined directly in the encoder embedding space, so the inference network parameterizes latent structure within that space rather than in an additional projected representation.

**EMA target encoder.** When stop-gradient is replaced by an exponential moving average (EMA) target encoder, the target embeddings $\mathbf{z}_j$ are produced by a separate copy of the encoder whose parameters $\xi$ are updated as $\xi \leftarrow m\,\xi + (1-m)\,\theta$ after each optimizer step, with the momentum coefficient $m$ following a cosine schedule from an initial value to 1.0. The EMA encoder is used in inference mode only (no gradient computation), and its parameters are not part of the optimization. This provides the same fixed-observation semantics as stop-gradient while smoothing the target representations across training steps.

**Latent sampling and likelihood evaluation.** A latent sample $\mathbf{s}_i = \boldsymbol{\mu}_i + \boldsymbol{\sigma}_i \odot \boldsymbol{\varepsilon}_i$, with $\boldsymbol{\varepsilon}_i \sim \mathcal{N}(\mathbf{0}, \mathbf{I})$, is drawn using the reparameterization trick (Kingma and Welling, 2014; Rezende et al., 2014). In practice, we use a single reparameterized sample ($K{=}1$) per view for all reported experiments; Appendix C.2 confirms that increasing $K$ yields no measurable improvement. The shared degrees-of-freedom parameter $\nu > 0$ controls the tail heaviness of both the directional and radial Student–$t$ likelihood terms; its role is developed in Section 4 and its effect is studied empirically in Section 5.5. Defining $\hat{\mathbf{s}}_i = \mathbf{s}_i / \|\mathbf{s}_i\|$, the negative log-likelihood averages

directional and radial terms across both conditional directions $(i, j) \in \{(1, 2), (2, 1)\}$:

$$\mathcal{L}_{\text{NLL}} = \tfrac{1}{2} \sum_{(i,j)\in\{(1,2),(2,1)\}} \mathbb{E}_{\mathbf{s}_i \sim q_i} \left[ \ell_{\text{dir}}\big(\text{sg}(\hat{\mathbf{z}}_j), \hat{\mathbf{s}}_i; \boldsymbol{\sigma}_i^2\big) + \ell_{\text{rad}}\big(\Delta r_{ij}\big) \right], \tag{4}$$

where $\text{sg}(\cdot)$ denotes a stop-gradient (or equivalently the detached EMA target) and terms constant with respect to $\theta, \phi$ are omitted for clarity. The *directional* term $\ell_{\text{dir}}$ is a Student–$t$ negative log-likelihood on the unit sphere, constructed via the geodesic log-map (do Carmo, 1992), a Schur-complement restriction of $\Sigma$ to the tangent space (Zhang, 2005), and the exponential-map Jacobian (do Carmo, 1992; Mardia and Jupp, 2000) (derived in Section 4.2):

$$\ell_{\text{dir}} = \tfrac{\nu+k}{2} \log\Big(1 + \tfrac{Q_{\text{dir}}}{\nu}\Big) + \tfrac{1}{2}\log|\Sigma_{\text{tan}}| + \log|J_{\text{exp}}|, \tag{5}$$

where $k = D-1$ is the tangent-space dimension. The first term scores the angular discrepancy between $\hat{\mathbf{s}}_i$ and $\hat{\mathbf{z}}_j$ under anisotropic whitening restricted to the tangent space at $\hat{\mathbf{s}}_i$, with $Q_{\text{dir}}$ denoting the Schur-complement Mahalanobis distance that properly accounts for the constraint $\mathbf{t} \perp \hat{\mathbf{s}}_i$. The second term penalizes variance inflation through the log-determinant of the restricted tangent-space covariance $\Sigma_{\text{tan}}$. The third term is the exponential-map Jacobian $\log|J_{\text{exp}}| = (D-2)(\log\sin\theta - \log\theta) \leq 0$, which corrects for the curvature of $\mathbb{S}^{D-1}$. The complete derivations are given in Section 4.2.

The *radial* term $\ell_{\text{rad}}$ is a one-dimensional Student–$t$ negative log-likelihood acting on the norm residual:

$$\Delta r_{ij} = \|\mathbf{z}_j\| - \|\mathbf{s}_i\|, \qquad \ell_{\text{rad}} = \tfrac{\nu+1}{2}\log\Big(1 + \tfrac{\Delta r_{ij}^2}{\nu}\Big). \tag{6}$$

**KL regularization and total objective.** To anchor the posteriors, each $q_i(\mathbf{s})$ is regularized toward a standard Gaussian prior $p(\mathbf{s}) = \mathcal{N}(\mathbf{0}, \mathbf{I})$ using the analytic KL divergence:

$$\mathcal{L}_{\text{KL}} = \frac{1}{2} \sum_{i=1}^{2} \sum_{d=1}^{D} \Big( \sigma_{i,d}^2 + \mu_{i,d}^2 - 1 - \log\sigma_{i,d}^2 \Big). \tag{7}$$

The final training loss combines the likelihood and regularization terms with a weighting factor $\beta$:

$$\mathcal{L} = \mathcal{L}_{\text{NLL}} + \beta\,\mathcal{L}_{\text{KL}}. \tag{8}$$

Appendix A provides pseudocode for the forward pass and loss computation of VJE, as well as additional implementation details to assist with reproducibility.

## 4 Latent variable model

We begin our theoretical formulation by defining a likelihood model $p_\psi(\mathbf{z} \mid \mathbf{s})$ that formalizes the probabilistic relationship between the latent variable $\mathbf{s}$ and the observed embedding $\mathbf{z}$, both $D$-dimensional vectors in $\mathbb{R}^D$. Here, $\mathbf{z}$ is produced by the encoder and $\mathbf{s}$ is sampled from a variational posterior $q(\mathbf{s} \mid \mathbf{z})$, with the two corresponding to different views of the same input. The variational posterior is the inference model, parameterized by the inference network defined in Section 3; the likelihood is the generative scoring model developed in this section. The likelihood evaluates how well a latent representation inferred from one view explains the embedding observed from another view. We ultimately evaluate the likelihood on the observation $y = (\hat{\mathbf{z}}, \|\mathbf{z}\|)$ introduced in Section 3, and write the final model as $p_\psi(y \mid \mathbf{s})$. This construction provides the foundation for our conditional evidence lower bound (ELBO) objective.

Our approach makes several explicit modelling choices that are motivated by geometric and statistical considerations. These choices are developed throughout this section and empirically evaluated in Section 5. The distinction between the likelihood-based formulation presented here and pointwise energy-based objectives is formalized at the objective level in Appendix B.

**Likelihood distribution.** The choice of likelihood distribution $p_\psi(\mathbf{z} \mid \mathbf{s})$ is a central design element, as it determines how residuals between $\mathbf{z}$ and $\mathbf{s}$ are scored under a normalized density and how sensitive the resulting objective is to large deviations. To motivate the final form we adopt, we first examine the behaviour of a Gaussian likelihood and its limitations. Choosing a Gaussian likelihood yields:

$$p_\psi^{\mathcal{N}}(\mathbf{z} \mid \mathbf{s}) = (2\pi\lambda)^{-D/2} \exp\left(-\tfrac{1}{2\lambda}\|\mathbf{z} - \mathbf{s}\|^2\right), \tag{9}$$

with corresponding negative log-likelihood (NLL) and gradient:

$$\ell_{\mathcal{N}}(\mathbf{z}; \mathbf{s}) = \tfrac{1}{2\lambda}\|\mathbf{z} - \mathbf{s}\|^2, \qquad \nabla_{\mathbf{z}}\ell_{\mathcal{N}} = \tfrac{1}{\lambda}(\mathbf{z} - \mathbf{s}). \tag{10}$$

The gradient norm $\|\nabla_{\mathbf{z}}\ell_{\mathcal{N}}\|$ grows linearly with $\|\mathbf{z} - \mathbf{s}\|$, leading to unbounded influence from large residuals.

Since the Gaussian loss grows quadratically with $\|\mathbf{z} - \mathbf{s}\|$, large deviations in high-dimensional spaces can dominate training dynamics. To mitigate this behaviour in a model-based manner, we instead adopt a heavy-tailed Student–$t$ likelihood. The Student–$t$ likelihood yields bounded influence in the corresponding NLL gradients (Huber and Ronchetti, 2009), ensuring that large residuals continue to contribute to the objective without exerting unbounded influence on gradients or parameter estimates. This robustness arises from the probabilistic form of the likelihood itself, rather than from ad-hoc clipping or heuristic penalties. The resulting likelihood is given by (Kotz and Nadarajah, 2004; Bishop, 2006):

$$p_\psi^{t,\Sigma}(\mathbf{z} \mid \mathbf{s}) = C_{\nu,D}\,|\Sigma|^{-1/2}\left[1 + \tfrac{1}{\nu}(\mathbf{z} - \mathbf{s})^\top\Sigma^{-1}(\mathbf{z} - \mathbf{s})\right]^{-\frac{\nu+D}{2}}, \tag{11}$$

yielding NLL and gradient functions:

$$\ell_{t,\Sigma}(\mathbf{z}; \mathbf{s}) = \tfrac{\nu+D}{2}\log\left(1 + \tfrac{1}{\nu}(\mathbf{z} - \mathbf{s})^\top\Sigma^{-1}(\mathbf{z} - \mathbf{s})\right) + \tfrac{1}{2}\log|\Sigma|,$$

$$\nabla_{\mathbf{z}}\ell_{t,\Sigma} = \frac{\nu + D}{\nu + (\mathbf{z} - \mathbf{s})^\top\Sigma^{-1}(\mathbf{z} - \mathbf{s})}\,\Sigma^{-1}(\mathbf{z} - \mathbf{s}), \tag{12}$$

where $\nu > 0$ controls tail heaviness, $\Sigma$ is a symmetric positive-definite scale matrix, and $C_{\nu,D}$ is the normalizing constant. As $\nu \to \infty$, the Student–$t$ likelihood recovers the Gaussian case. Throughout this paper, all references to a "Gaussian" likelihood or the "Gaussian limit" refer to the $\nu \to \infty$ limit within the same Student–$t$ family, not to a separate unfactorized formulation.

This formulation yields an NLL that grows only logarithmically for large residuals, while the gradient magnitude decays with the Mahalanobis distance $(\mathbf{z} - \mathbf{s})^\top\Sigma^{-1}(\mathbf{z} - \mathbf{s})$. Figure 2 illustrates this behaviour, highlighting the bounded influence of the Student–$t$ likelihood compared to the unbounded Gaussian case.

Although the Student–$t$ likelihood provides robustness to large deviations, the Mahalanobis term $(\mathbf{z} - \mathbf{s})^\top\Sigma^{-1}(\mathbf{z} - \mathbf{s})$ still couples angular misalignment and differences in norms into a single error channel. Expanding this term,

$$(\mathbf{z} - \mathbf{s})^\top\Sigma^{-1}(\mathbf{z} - \mathbf{s}) = \mathbf{z}^\top\Sigma^{-1}\mathbf{z} + \mathbf{s}^\top\Sigma^{-1}\mathbf{s} - 2\,\mathbf{z}^\top\Sigma^{-1}\mathbf{s}, \tag{13}$$

reveals both the individual quadratic norms $\mathbf{z}^\top\Sigma^{-1}\mathbf{z}$, $\mathbf{s}^\top\Sigma^{-1}\mathbf{s}$ and their inner product $\mathbf{z}^\top\Sigma^{-1}\mathbf{s}$. As a consequence, the contribution of an angular discrepancy between $\mathbf{z}$ and $\mathbf{s}$ is scaled by their norms (in the $\Sigma^{-1}$ metric), so that large-norm embeddings can produce large Mahalanobis residuals even for moderate angular error. In contrast, a large Mahalanobis residual does not reveal whether it is dominated by a mismatch in direction or magnitude. This coupling between scale and orientation motivates a reparameterization in which directional and radial contributions are modelled by separate likelihood factors.

To do so, we reformulate the likelihood in a space where angular and radial variations are explicitly decoupled, so that each embedding is represented not by its Euclidean coordinates but by its *direction* and *magnitude*. In the radial channel, discrepancies are evaluated relative to the predicted norm via the residual:

$$\Delta r := \|\mathbf{z}\| - \|\mathbf{s}\| \in \mathbb{R}. \tag{14}$$

This representation allows us to treat directional and radial channels separately in the likelihood, each with its own stability properties and normalization behaviour. We define a product-form likelihood on the observation $y = (\hat{\mathbf{z}}, \|\mathbf{z}\|)$, with the radial factor expressed in terms of the residual $\Delta r$.

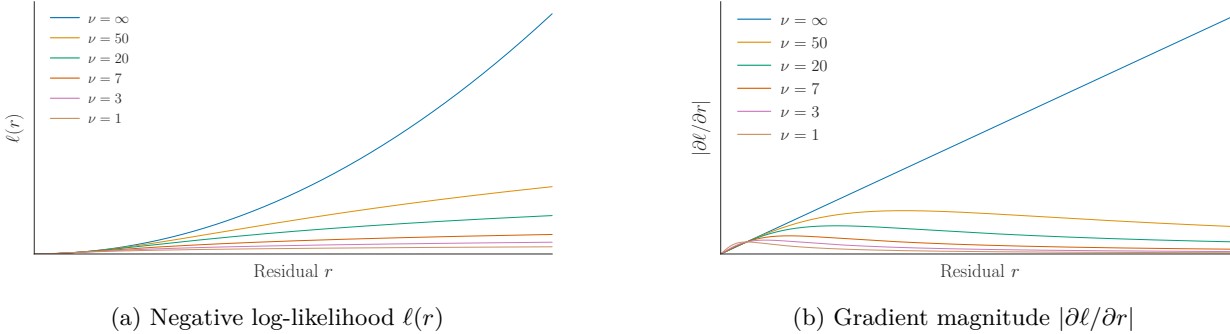

(a) Negative log-likelihood $\ell(r)$          (b) Gradient magnitude $|\partial\ell/\partial r|$

Figure 2: One-dimensional scalar illustration of the Student–$t$ versus Gaussian negative log-likelihood and gradient magnitude, plotted as functions of a residual $r$ at fixed scale $\lambda = 1$ for different degrees of freedom $\nu$ (with the Gaussian limit at $\nu = \infty$). Panel (a) illustrates how heavy tails moderate the growth of the negative log-likelihood for large residuals, while panel (b) shows the corresponding influence functions, where gradients saturate and then decay so that outliers contribute only a bounded amount of signal. This underlines the choice of Student–$t$ likelihoods in VJE to stabilize training without ad-hoc heuristics. Note that this figure illustrates the qualitative behaviour of the likelihood family; the full training objective uses the factorized directional–radial form developed in Sections 4.1–4.3.

In Section 4.1, we develop this decomposition by examining how a polar factorization of the isotropic Student–$t$ distribution motivates the separation of directional and radial channels. In Section 4.2, we introduce feature-wise (i.e., anisotropic) uncertainty in the directional term on the unit sphere. We first present a normalized spherical reference model, and then derive the directional subdensity used in training by omitting a normalization contribution that proved empirically unstable. Section 4.3 formalizes the radial residual $\Delta r$ and its associated likelihood, and Section 4.4 establishes the conditional evidence lower bound (ELBO) that combines these components into a symmetric objective suitable for joint embedding architectures.

## 4.1 Polar decomposition of the likelihood

To motivate the separation of directional and radial channels, we begin from an isotropic Student–$t$ likelihood, whose rotational symmetry admits a well-defined polar decomposition into independent radial and directional factors (Mardia and Jupp, 2000). The isotropic Student–$t$ likelihood can be written as:

$$p_\psi^{t,\lambda}(\mathbf{z} \mid \mathbf{s}) = C_{\nu,D} \, \lambda^{-D/2} \left[ 1 + \frac{\|\mathbf{z} - \mathbf{s}\|^2}{\nu\lambda} \right]^{-\frac{\nu+D}{2}}, \tag{15}$$

where $\lambda > 0$ is a scalar scale parameter. This corresponds to the isotropic special case $\Sigma = \lambda\mathbf{I}$ of the elliptical Student–$t$ likelihood in Eq. (11).

We introduce polar coordinates for the displacement vector $\mathbf{z} - \mathbf{s}$:

$$\rho := \|\mathbf{z} - \mathbf{s}\| \in (0,\infty), \qquad \boldsymbol{\omega} := \frac{\mathbf{z} - \mathbf{s}}{\|\mathbf{z} - \mathbf{s}\|} \in \mathbb{S}^{D-1}, \tag{16}$$

and the corresponding Jacobian for this change of variables:

$$d\mathbf{z} = \rho^{D-1} \, d\rho \, d\boldsymbol{\omega}, \tag{17}$$

where $d\boldsymbol{\omega}$ denotes the uniform surface measure on the unit sphere $\mathbb{S}^{D-1}$ (Mardia and Jupp, 2000). Since the isotropic density in Eq. (15) depends only on $\rho$, the direction $\boldsymbol{\omega}$ is uniformly distributed and independent of $\rho$. The likelihood therefore factorizes into independent radial and directional components:

$$p_\psi^{t,\lambda}(\mathbf{z} \mid \mathbf{s}) = p_{\text{rad}}(\rho) \, p_{\text{dir}}(\boldsymbol{\omega}), \qquad p_{\text{dir}}(\boldsymbol{\omega}) = \frac{1}{\text{vol}(\mathbb{S}^{D-1})}. \tag{18}$$

The explicit form of the radial factor is given by:

$$p_{\text{rad}}(\rho) = \frac{2\,\Gamma\!\left(\frac{\nu+D}{2}\right)}{\Gamma\!\left(\frac{\nu}{2}\right)\Gamma\!\left(\frac{D}{2}\right)(\nu\lambda)^{D/2}}\,\rho^{D-1}\left[1+\frac{\rho^2}{\nu\lambda}\right]^{-\frac{\nu+D}{2}}, \qquad \rho > 0, \tag{19}$$

which provides a clear geometric interpretation, as the isotropic Student–$t$ distributes mass uniformly over directions, and the entire radial structure is captured by the corresponding one-dimensional term.

This polar decomposition motivates our use of a product-form likelihood (i.e., joint likelihood) with independent directional and radial factors, enabling separate treatment of angular and radial errors. While the isotropic case yields a well-defined factorization, our full model extends this structure to incorporate anisotropic scaling and alternative radial parameterizations. This design choice provides flexibility to model uncertainty while retaining probabilistic and geometric coherence. The specific parameterization of these factors is developed in subsequent sections.

### 4.2 Directional likelihood and feature-wise uncertainty

An immediate limitation of the isotropic form derived in Section 4.1 is the shared scale $\lambda$, which prevents the likelihood from expressing feature-wise variation. We enable anisotropic weighting of dimensions by introducing a diagonal variance vector $\boldsymbol{\sigma}^2 \in \mathbb{R}_+^D$ in the directional term. In the polar parametrization of the displacement $\mathbf{z} - \mathbf{s}$, the radial variable $\rho = \|\mathbf{z} - \mathbf{s}\|$ is one-dimensional, so any anisotropy is naturally confined to the directional term. We define the directional scale matrix as $\Sigma = \text{diag}(\boldsymbol{\sigma}^2)$.

Since both the target direction $\hat{\mathbf{z}}$ and predicted direction $\hat{\mathbf{s}}$ lie on $\mathbb{S}^{D-1}$, the directional likelihood should respect the geometry of the sphere. We therefore begin from a normalized spherical reference construction obtained by mapping $\hat{\mathbf{z}}$ into the tangent space at $\hat{\mathbf{s}}$, evaluating a Student–$t$ density in that $(D-1)$-dimensional space, and accounting for curvature through the exponential-map Jacobian. The stable training objective used in this work is then obtained from this reference construction by omitting a normalization contribution whose gradients were empirically unstable and promoted collapse, yielding a directional subdensity on $\mathbb{S}^{D-1}$.

**Geodesic log-map.** Given unit vectors $\hat{\mathbf{z}}, \hat{\mathbf{s}} \in \mathbb{S}^{D-1}$, let $\theta = \arccos(\hat{\mathbf{z}}^\top \hat{\mathbf{s}}) \in [0, \pi)$ denote their geodesic distance. The logarithmic map at $\hat{\mathbf{s}}$ sends $\hat{\mathbf{z}}$ to the tangent vector (do Carmo, 1992):

$$\mathbf{t} = \log_{\hat{\mathbf{s}}}(\hat{\mathbf{z}}) = \frac{\theta}{\sin\theta}\big(\hat{\mathbf{z}} - \cos\theta\,\hat{\mathbf{s}}\big) \in T_{\hat{\mathbf{s}}}\mathbb{S}^{D-1}, \tag{20}$$

which satisfies $\mathbf{t}^\top \hat{\mathbf{s}} = 0$ and $\|\mathbf{t}\| = \theta$. When $\theta \to 0$, the map reduces to $\mathbf{t} \to \hat{\mathbf{z}} - \hat{\mathbf{s}}$.

**Tangent-space covariance via Schur complement.** The ambient diagonal covariance $\Sigma = \text{diag}(\boldsymbol{\sigma}^2)$ must be restricted to the $(D-1)$-dimensional tangent space $T_{\hat{\mathbf{s}}}\mathbb{S}^{D-1}$, which is the orthogonal complement of the normal $\mathbf{n} = \hat{\mathbf{s}}$. Following the Schur-complement construction for constrained quadratic forms (Zhang, 2005), the Mahalanobis distance restricted to the tangent space is:

$$Q_{\text{dir}} = \mathbf{t}^\top \Sigma^{-1} \mathbf{t} - \frac{(\mathbf{t}^\top \Sigma^{-1}\mathbf{n})^2}{\mathbf{n}^\top \Sigma^{-1}\mathbf{n}}, \tag{21}$$

where the subtracted term removes the component of $\Sigma^{-1}\mathbf{t}$ along $\mathbf{n}$. Although $\mathbf{t} \perp \mathbf{n}$ in the Euclidean metric, the whitened vector $\Sigma^{-1}\mathbf{t}$ generically has a nonzero projection onto $\mathbf{n}$ under $\Sigma^{-1}$, which the Schur complement corrects.

The determinant of the restricted $(D-1)$-dimensional tangent-space covariance is (Zhang, 2005):

$$\det(\Sigma_{\text{tan}}) = \det(\Sigma) \cdot (\mathbf{n}^\top \Sigma^{-1}\mathbf{n}) = \left(\prod_{d=1}^{D}\sigma_d^2\right)\cdot\left(\sum_{d=1}^{D}\frac{\hat{s}_d^2}{\sigma_d^2}\right), \tag{22}$$

so that the tangent-space log-determinant is:

$$\tfrac{1}{2}\log\det(\Sigma_{\text{tan}}) = \tfrac{1}{2}\sum_{d=1}^{D}\log\sigma_d^2 + \tfrac{1}{2}\log\!\left(\sum_{d=1}^{D}\frac{\hat{s}_d^2}{\sigma_d^2}\right). \tag{23}$$

**Exponential-map Jacobian.** The exponential map $\exp_{\hat{\mathbf{s}}} : T_{\hat{\mathbf{s}}}\mathbb{S}^{D-1} \to \mathbb{S}^{D-1}$ maps tangent vectors to the sphere. In geodesic normal coordinates at distance $\theta$ from the base point, the volume element on $\mathbb{S}^{D-1}$ relative to the flat tangent space satisfies (do Carmo, 1992; Mardia and Jupp, 2000):

$$\frac{d\text{vol}_{\mathbb{S}}}{d\text{vol}_T} = \left(\frac{\sin\theta}{\theta}\right)^{D-2}. \tag{24}$$

A tangent-space density $q_{\text{tan}}(\mathbf{t})$ therefore induces a density on the sphere via $p_{\mathbb{S}}(\hat{\mathbf{z}}) = q_{\text{tan}}(\mathbf{t}) \cdot (\theta/\sin\theta)^{D-2}$, and the corresponding log-Jacobian correction entering the NLL is:

$$\log|J_{\exp}| = (D-2)(\log\sin\theta - \log\theta) \le 0. \tag{25}$$

**Normalized reference construction and directional subdensity.** Combining the tangent-space Student–$t$ with the restricted covariance and the Jacobian correction yields the normalized spherical reference construction

$$p_{\text{dir}}^{\text{ref}}(\hat{\mathbf{z}} \mid \hat{\mathbf{s}}, \boldsymbol{\sigma}^2) \propto C_{\nu,k} \det(\Sigma_{\text{tan}})^{-1/2} \left(1 + \tfrac{1}{\nu}Q_{\text{dir}}\right)^{-\frac{\nu+k}{2}} \left(\tfrac{\theta}{\sin\theta}\right)^{D-2}, \qquad k = D-1, \tag{26}$$

where the omitted proportionality constant is the parameter-independent normalization over $\mathbb{S}^{D-1}$. Direct optimization of the corresponding fully normalized form introduces a normalization contribution whose gradients were empirically unstable and consistently promoted collapse. We therefore omit this contribution and use the resulting directional subdensity in training. Dropping constants independent of $(\boldsymbol{\mu}, \boldsymbol{\sigma}^2)$, the per-sample directional negative log-likelihood is the expression given in Eq. (5), restated here:

$$\ell_{\text{dir}} = \frac{\nu+k}{2}\log\left(1 + \tfrac{1}{\nu}Q_{\text{dir}}\right) + \tfrac{1}{2}\log\det(\Sigma_{\text{tan}}) + (D-2)(\log\sin\theta - \log\theta), \tag{27}$$

where $k = D-1$ is the tangent-space dimension. The relationship to the directional subdensity is

$$\ell_{\text{dir}} = -\log\tilde{p}_{\text{dir}}(\hat{\mathbf{z}} \mid \hat{\mathbf{s}}, \boldsymbol{\sigma}^2) + \text{const}, \tag{28}$$

where $\tilde{p}_{\text{dir}}$ denotes the subnormalized directional density and the constant absorbs the Student–$t$ normalization $C_{\nu,k}$. The three terms play complementary roles: the first scores the angular discrepancy under anisotropic whitening with heavy-tailed robustness; the second penalizes variance inflation; and the third accounts for the curvature of $\mathbb{S}^{D-1}$.

Because $\tilde{p}_{\text{dir}}$ is nonnegative and subnormalized on the sphere, it defines a valid directional likelihood contribution within the associated subprobability model used in training. This is the directional construction used throughout all reported experiments.

**Variance tying.** To ensure that feature-wise uncertainty jointly governs the directional likelihood and the variational posterior, we tie the variance vector $\boldsymbol{\sigma}^2$ between the two distributions. The inference network output $\boldsymbol{\sigma}^2$ is shared across the Gaussian posterior $q(\mathbf{s} \mid \mathbf{z})$ and the directional likelihood, with the same matrix $\Sigma = \text{diag}(\boldsymbol{\sigma}^2)$ appearing in both. This choice makes each $\sigma_d^2$ a shared per-feature scale parameter: it controls the weighting of the directional residual in the likelihood while also serving as the variance parameter of the posterior, so that feature-wise uncertainty is expressed consistently across both distributions.

### 4.3 Radial reparameterization and final likelihood

While the polar factorization in Section 4.1 yields a one-dimensional radial variable $\rho = \|\mathbf{z} - \mathbf{s}\|$, using this Euclidean distance directly as the radial term still couples angular misalignment with differences in norms. The squared distance expansion

$$\|\mathbf{z} - \mathbf{s}\|^2 = \|\mathbf{z}\|^2 + \|\mathbf{s}\|^2 - 2\|\mathbf{z}\|\|\mathbf{s}\|\cos\theta \tag{29}$$

reveals that the Euclidean distance inherently couples angular alignment with magnitude through the cosine term, where $\theta$ denotes the angle between $\mathbf{z}$ and $\mathbf{s}$. Angular discrepancies therefore contribute to the radial error in proportion to the product of norms. For a Student-$t$ likelihood, the angular gradient takes the form:

$$\frac{\partial\ell_t}{\partial\theta} = \frac{\nu+D}{\nu\lambda + \|\mathbf{z} - \mathbf{s}\|^2}\|\mathbf{z}\|\|\mathbf{s}\|\sin\theta, \tag{30}$$

which remains susceptible to norm amplification despite the bounded prefactor.

To address this coupling, we reparameterize the radial channel as the *difference of norms*:

$$\Delta r \; := \; \|\mathbf{z}\| - \|\mathbf{s}\|. \tag{31}$$

This parameterization measures magnitude discrepancy independently of angular alignment: the radial residual is zero when predicted and observed magnitudes agree, regardless of their relative directions. The reference point shifts from the Euclidean distance $\|\mathbf{z} - \mathbf{s}\|$ to the predicted norm $\|\mathbf{s}\|$, enabling a clean separation of scale from orientation in the radial term.

Consequently, translation invariance is no longer preserved, since $\|(\mathbf{z} + \mathbf{a}) - (\mathbf{s} + \mathbf{a})\| = \|\mathbf{z} - \mathbf{s}\|$ but $\|\mathbf{z} + \mathbf{a}\| - \|\mathbf{s} + \mathbf{a}\| \neq \|\mathbf{z}\| - \|\mathbf{s}\|$. However, this does not introduce a geometric inconsistency within the present model, as we anchor both the posterior $q(\mathbf{s} \mid \mathbf{z})$ and the likelihood $p_\psi(\mathbf{z} \mid \mathbf{s})$ to the origin by introducing the standard Gaussian prior in Section 4.4. Since both the likelihood and the posterior are defined relative to this fixed coordinate system, the reparameterization remains geometrically consistent with the choice of a fixed origin.

The resulting radial likelihood is given by a one-dimensional Student-$t$ kernel:

$$p_{\text{rad}}^{(\Delta)}(\Delta r) \; = \; \frac{\Gamma(\frac{\nu+1}{2})}{\sqrt{\nu \pi \lambda}\, \Gamma(\frac{\nu}{2})} \left(1 + \frac{(\Delta r)^2}{\nu \lambda}\right)^{-\frac{\nu+1}{2}}, \qquad \Delta r \in \mathbb{R}. \tag{32}$$

Therefore the per-sample NLL, omitting constants, is:

$$\ell_{\text{rad}}(\Delta r) = \frac{\nu+1}{2} \log\left(1 + \frac{\Delta r^2}{\nu \lambda}\right), \tag{33}$$

with the derivative:

$$\left| \frac{\partial}{\partial \Delta r} \left[ -\log p_{\text{rad}}^{(\Delta)}(\Delta r) \right] \right| \; = \; \frac{\nu+1}{\nu \lambda} \frac{|\Delta r|}{1 + (\Delta r)^2/(\nu \lambda)} \; \leq \; \frac{\nu+1}{2\sqrt{\nu \lambda}}, \tag{34}$$

which is bounded for any finite $\nu$. In the radial kernel $p_{\text{rad}}^{(\Delta)}(\Delta r)$, the parameter $\lambda > 0$ acts purely as a global scale: the density depends on $\Delta r$ only through the combination $\Delta r^2/(\nu \lambda)$. A change in $\lambda$ is therefore equivalent to a rescaling of the radial coordinate and does not increase the expressiveness of the model. Moreover, for any fixed $\Delta r \neq 0$ and $\nu > 0$, increasing $\lambda$ decreases the radial NLL $\ell_{\text{rad}}(\Delta r)$. An unconstrained maximum-likelihood solution would therefore push $\lambda \to \infty$, effectively eliminating the radial penalty. To avoid this degenerate path, we fix the radial scale to $\lambda = 1$, and leave $\nu$ as the sole shared radial hyperparameter.

Combining the stabilized radial term (32) with the directional factor of Eq. (27) yields the factorized conditional likelihood used in training on the observation $y = (\hat{\mathbf{z}}, \|\mathbf{z}\|)$:

$$\tilde{p}_\psi(y \mid \mathbf{s}, \boldsymbol{\sigma}^2) \; = \; p_{\text{rad}}^{(\Delta)}(\Delta r) \cdot \tilde{p}_{\text{dir}}(\hat{\mathbf{z}} \mid \hat{\mathbf{s}}, \boldsymbol{\sigma}^2), \qquad \Delta r := \|\mathbf{z}\| - \|\mathbf{s}\|. \tag{35}$$

Constants independent of $(\boldsymbol{\mu}, \boldsymbol{\sigma}^2)$ are omitted from the loss. The Jacobian terms associated with the change of variables from $\mathbf{z}$ to $(\hat{\mathbf{z}}, \|\mathbf{z}\|)$ do not depend on the trainable parameters and are absorbed into these omitted constants.

Together, the radial factor derived in this section and the directional factor of Eq. (27) specify the complete subnormalized conditional model $\tilde{p}_\psi(y \mid \mathbf{s}, \boldsymbol{\sigma}^2)$ on representation-space observations.

## 4.4 Conditional evidence lower bound

To establish the evidence lower bound (ELBO) underlying the VJE objective, we combine the factorized conditional model $\tilde{p}_\psi(y \mid \mathbf{s}, \boldsymbol{\sigma}^2)$ of Eq. (35) with the diagonal-Gaussian variational posterior and KL divergence introduced in Section 3 (Eqs. (4)–(7)).

For each conditional direction $i \rightarrow j$ with $(i, j) \in \{(1, 2), (2, 1)\}$, the target embedding $\mathbf{z}_j$ is treated as a fixed observation by detaching it from the computation graph, so that $y_j$ appears only as an observed variable in the likelihood term. The one-way conditional ELBO is then:

$$\mathcal{F}_{i \rightarrow j} = \mathbb{E}_{\mathbf{s}_i \sim q_i} \left[ \log p_{\mathrm{rad}}^{(\Delta)}(\|\mathbf{z}_j\| - \|\mathbf{s}_i\|) + \log \tilde{p}_{\mathrm{dir}}(\hat{\mathbf{z}}_j \mid \hat{\mathbf{s}}_i, \boldsymbol{\sigma}_i^2) \right] - \mathcal{L}_{\mathrm{KL}}(q_i \| p), \qquad (36)$$

which provides the corresponding variational lower bound on the conditional log-mass of the associated subprobability model

$$\log \tilde{p}_\psi(y_j \mid \mathbf{z}_i) \ \geq \ \mathcal{F}_{i \rightarrow j}, \qquad (37)$$

where $\tilde{p}_\psi$ is the factorized subnormalized conditional model defined in Eq. (35). Since the Jacobian of the mapping from $\mathbf{z}_j$ to $(\hat{\mathbf{z}}_j, \|\mathbf{z}_j\|)$ does not depend on the model parameters $(\boldsymbol{\mu}_i, \boldsymbol{\sigma}_i^2)$, the same bound holds up to an additive constant that is omitted from the loss.

Symmetrizing over both directions yields the $\beta$-weighted objective:

$$\overline{\mathcal{F}}^{(\beta)} = \frac{1}{2} \sum_{(i,j) \in \{(1,2),(2,1)\}} \left\{ \mathbb{E}_{\mathbf{s}_i \sim q_i} \left[ \log p_{\mathrm{rad}}^{(\Delta)}(\|\mathbf{z}_j\| - \|\mathbf{s}_i\|) + \log \tilde{p}_{\mathrm{dir}}(\hat{\mathbf{z}}_j \mid \hat{\mathbf{s}}_i, \boldsymbol{\sigma}_i^2) \right] - \beta \, \mathcal{L}_{\mathrm{KL}}(q_i \| p) \right\}. \qquad (38)$$

Here, $\beta$ weights the KL regularization analogously to $\beta$-VAE (Higgins et al., 2017): $\beta = 1$ corresponds to the unweighted conditional ELBO, $\beta < 1$ emphasizes likelihood fitting, and $\beta > 1$ increases regularization strength.

For training, we minimize the negative of Eq. (38). Writing the likelihood terms as their negative log-likelihoods recovers the training loss $\mathcal{L} = \mathcal{L}_{\mathrm{NLL}} + \beta \, \mathcal{L}_{\mathrm{KL}}$ of Eq. (8), with $\mathcal{L}_{\mathrm{NLL}}$ as in Eq. (4) and $\mathcal{L}_{\mathrm{KL}}$ as in Eq. (7). The target $\mathbf{z}_j$ is always detached, ensuring that gradients flow only through the inference network of branch $i$, consistent with the fixed-observation semantics.

**Joint likelihood at inference.** At test time, the factorized likelihood of Eq. (35) yields a per-sample scoring function. Given a single input $\mathbf{x}$, we encode $\mathbf{z} = f_\theta(\mathbf{x})$, infer $(\boldsymbol{\mu}, \boldsymbol{\sigma}^2) = g_\phi(\mathbf{z})$, and set $\mathbf{s} = \boldsymbol{\mu}$. The joint likelihood score is the resulting negative log-likelihood (NLL):

$$S(\mathbf{x}) = \ell_{\mathrm{dir}}(\hat{\mathbf{z}}, \hat{\boldsymbol{\mu}}; \boldsymbol{\sigma}^2) + \ell_{\mathrm{rad}}(\|\mathbf{z}\| - \|\boldsymbol{\mu}\|) \qquad (39)$$

This score requires no auxiliary data or multiple views, with lower values indicating better fit under the model. In Section 5.4, we evaluate this score for out-of-distribution detection as a downstream assessment of the learned uncertainty.

# 5 Experiments

We evaluate the proposed Variational Joint Embedding (VJE) framework along three complementary axes. First, we assess whether VJE retains discriminative representation quality comparable to deterministic joint-embedding baselines, establishing that the variational objective supports competitive feature learning (Section 5.2). Second, we study the geometry of the learned representations and posterior structure to determine whether the model's probabilistic quantities organize meaningfully over representation space rather than behaving as unstructured auxiliary outputs (Section 5.3). Third, we assess whether these probabilistic quantities exhibit coherent semantics in a downstream task, using out-of-distribution detection as a testbed (Section 5.4). Additionally, we perform ablations to quantify the contributions of key design choices, including tail heaviness and the individual loss components (Section 5.5), and conclude with a brief synthesis of the main empirical findings (Section 5.6). Implementation details are provided in Appendix A, while additional supporting experiments, ablations, and diagnostics are presented in Appendix C.

## 5.1 Experimental setup

All experiments use a ResNet backbone (He et al., 2016) for consistency, followed by an inference network that outputs posterior parameters $(\boldsymbol{\mu}, \boldsymbol{\sigma}^2)$ as described in Section 3. For ImageNet–1K (Russakovsky et al., 2015)

and the supplementary ImageNet–100 experiments reported in the appendices, we use a ResNet–50 encoder, while for CIFAR–10, CIFAR–100 (Krizhevsky, 2009), STL–10 (Coates et al., 2011), and Tiny ImageNet (Le and Yang, 2015), we use a smaller ResNet–18 model. For CIFAR–10 and CIFAR–100, the architecture is adapted to 32×32 inputs by changing the first convolution to 3×3 with stride 1 and removing the initial max-pooling layer. All models are trained from random initialization.

The inference network used in our experiments is a two-layer bottleneck MLP interleaved with Layer Normalization (Ba et al., 2016) and nonlinear activation, with bottleneck dimension 512 for ResNet–50 and 128 for ResNet–18. Two separate linear heads then produce the posterior mean $\boldsymbol{\mu}$ and diagonal variance $\boldsymbol{\sigma}^2$, exactly as described in Section 3 and Appendix A.1. VJE does not use a separate projection head, and the latent-variable model and conditional ELBO are therefore defined directly on encoder features.

Unless otherwise noted, we use $\beta{=}1.0$, $\nu{=}1.0$, and a single reparameterized sample per input. Appendix C.2 shows that increasing the number of Monte Carlo samples provides no measurable benefit in our setting. For representation learning on ImageNet–1K, we evaluate frozen representations through a linear probe, while on CIFAR–10, CIFAR–100, STL–10, Tiny ImageNet, and ImageNet–100, we report $k$–nearest-neighbour accuracy over the course of training with $k{=}20$.

## 5.2 Representation learning

**ImageNet–1K.** We begin with a large-scale evaluation on ImageNet–1K to assess whether the variational objective compromises discriminative capacity. Training follows a 100-epoch pretraining schedule with ResNet–50, an EMA target encoder, and the BYOL augmentation recipe (Grill et al., 2020). We use stochastic gradient descent with momentum 0.9, batch size 256, cosine learning-rate decay (Loshchilov and Hutter, 2017) from 0.05, weight decay $5{\times}10^{-4}$ with normalization layers and biases excluded, and a 10-epoch linear warm-up. After pretraining, we train a linear classifier on frozen backbone features for 100 epochs using stochastic gradient descent with momentum 0.9, a cosine learning-rate schedule, following standard linear evaluation practice in prior self-supervised learning work (Bardes et al., 2022; Chen and He, 2021).

Table 1: Linear evaluation on ImageNet–1K with ResNet–50 and 100 pretraining epochs. VJE uses an EMA target encoder. Mean and standard deviation for VJE are computed over three runs. Published baseline values are taken from the cited source corresponding to each method; entries marked (*) denote 100-epoch results reported by Chen and He (2021).

| Method | Top-1 (%) |
|---|---|
| SimCLR (Chen et al., 2020a) | 66.5* |
| BYOL (Grill et al., 2020) | 66.5* |
| SwAV (Caron et al., 2020) | 66.5 |
| SimSiam (Chen and He, 2021) | 68.1 |
| VICReg (Bardes et al., 2022) | 68.6 |
| **VJE (ours)** | $68.2 \pm 0.3$ |

VJE achieves $68.2 \pm 0.3\%$ top-1 accuracy on ImageNet–1K and $88.4 \pm 0.5\%$ top-5 accuracy, placing it in the same range as strong deterministic baselines under the same epoch budget (Table 1). These results indicate that the variational formulation does not compromise discriminative capacity, as the model remains competitive in representation learning despite the stochasticity introduced by the training objective.

**Multi-dataset comparison.** We next compare VJE against deterministic joint-embedding baselines on CIFAR–10, CIFAR–100, and STL–10. For controlled comparison, we train all methods with a shared ResNet–18 backbone for 800 epochs, following the long-horizon CIFAR evaluation regime commonly used for SimSiam-style comparisons (Chen and He, 2021). The deterministic baselines reported in this section are our reproductions under this shared encoder and epoch budget; method-specific choices such as head design, optimizer, learning-rate schedule, and augmentation recipe otherwise follow the corresponding published formulations (Chen and He, 2021; Bardes et al., 2022). In particular, SimSiam (Chen and He, 2021) uses

Table 2: $k$–NN accuracy (%, $k$=20) on three datasets with ResNet–18 after 800 pretraining epochs. All methods share the same encoder architecture and epoch budget. SimSiam and VICReg are reproduced under this common setup while retaining their method-specific published design choices (Chen and He, 2021; Bardes et al., 2022); VJE uses no projection head. Mean $\pm$ std over three runs.

| Method | | CIFAR–10 | CIFAR–100 | STL–10 |
|---|---|---|---|---|
| SimSiam | | $90.5 \pm 0.2$ | $53.2 \pm 0.5$ | $74.7 \pm 2.1$ |
| VICReg | | $86.4 \pm 0.1$ | $59.4 \pm 0.2$ | $82.9 \pm 0.1$ |
| VJE (stop-grad) | $z$ | $89.5 \pm 0.2$ | $59.7 \pm 0.4$ | $83.9 \pm 0.7$ |
| | $\mu$ | $89.5 \pm 0.2$ | $58.7 \pm 0.5$ | $83.3 \pm 0.8$ |
| VJE (EMA) | $z$ | $91.4 \pm 0.2$ | $63.0 \pm 0.3$ | $87.9 \pm 0.6$ |
| | $\mu$ | $91.4 \pm 0.2$ | $62.8 \pm 0.3$ | $87.5 \pm 0.6$ |
| | $\tilde{z}$ | $\mathbf{91.8 \pm 0.1}$ | $\mathbf{64.1 \pm 0.4}$ | $\mathbf{88.2 \pm 0.3}$ |

a two-layer projector with a bottleneck predictor, whereas VICReg (Bardes et al., 2022) uses a three-layer 4096-dimensional expander. VJE uses the BYOL augmentation recipe (Grill et al., 2020), stochastic gradient descent with momentum 0.9, batch size 256, cosine learning-rate decay from 0.05, weight decay $5 \times 10^{-4}$ (excluding normalization layers and biases), and a 10-epoch linear warm-up.

We report $k$–nearest-neighbour accuracy ($k$=20) on the test set, evaluating every 10 epochs. For VJE, we evaluate the encoder output $z$, the posterior mean $\mu$, and, for the EMA variant, the EMA encoder output $\tilde{z}$. Results are averaged over three runs.

Table 2 shows that VJE is competitive with or outperforms both reproduced deterministic baselines across all three datasets. On CIFAR–10, SimSiam is the stronger deterministic baseline, and VJE (stop-grad) remains within one point. On CIFAR–100 and STL–10, VJE (stop-grad) exceeds both baselines, while using an EMA target encoder yields further improvements across all datasets, with $\tilde{z}$ providing the strongest representation overall.

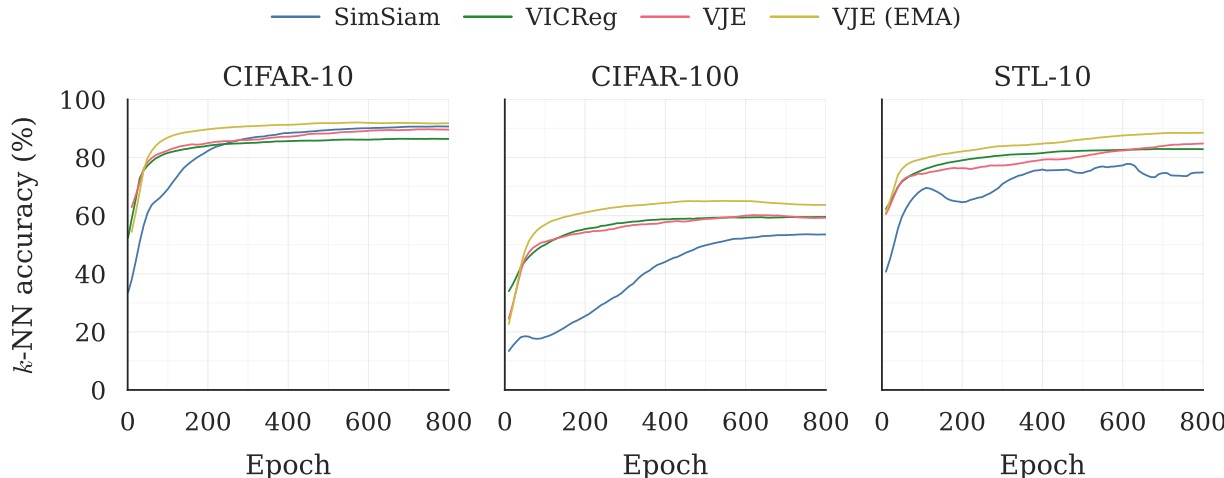

Figure 3: $k$–NN accuracy ($k$=20) over training on CIFAR–10 (left), CIFAR–100 (centre), and STL–10 (right). Curves show SimSiam, VICReg, VJE (stop-grad, $z$), and VJE (EMA, $\tilde{z}$). VJE (EMA) attains the highest plateau on all three datasets, and SimSiam exhibits visibly less stable training on STL–10.

The encoder output $z$ and posterior mean $\mu$ perform nearly identically on CIFAR–10 and remain close on CIFAR–100 and STL–10, where $\mu$ trails by only 0.5–1.0 points. This indicates that the posterior remains tightly centered around the encoder representation rather than displacing its semantic content.

Figure 3 is consistent with the endpoint results in Table 2: VJE (EMA) reaches the highest plateau on all three datasets, while SimSiam is visibly less stable on STL–10.

### 5.3 Geometry of learned uncertainty

To examine how the variational posterior is organized over the learned representation, we visualize the CIFAR–10 embedding together with posterior-derived statistics in Figure 4, and quantify the same structure directly in the original representation space in Table 3.

A first qualitative observation from Figure 4 is that, in the 2-D t-SNE layout, the posterior statistics exhibit coherent large-scale structure. Mean posterior variance and NLL show a broadly aligned pattern: the central, visually overlapping region tends to support higher uncertainty magnitude and worse fit, while more separated peripheral regions tend to show lower variance and lower NLL. In contrast, the KL term and the anisotropy of the posterior variance field, measured here as the coefficient of variation (CoV) of the per-dimension posterior variance $\boldsymbol{\sigma}^2$, display the opposite broad trend, with reduced values in the same central region and larger values away from it. This suggests that the posterior is not governed by a single scalar uncertainty field, but that different posterior-derived quantities respond differently across the representation.

This large-scale visual pattern is also consistent with the prior geometry built into the model. Since the KL term is taken with respect to the standard Gaussian prior $p(\mathbf{s}) = \mathcal{N}(\mathbf{0}, \mathbf{I})$ (Section 4.4), it measures the degree to which the posterior departs from an origin-anchored, unit-variance reference geometry. The broad low-KL central region in Figure 4 therefore corresponds to posteriors that remain closer to this prior anchor,

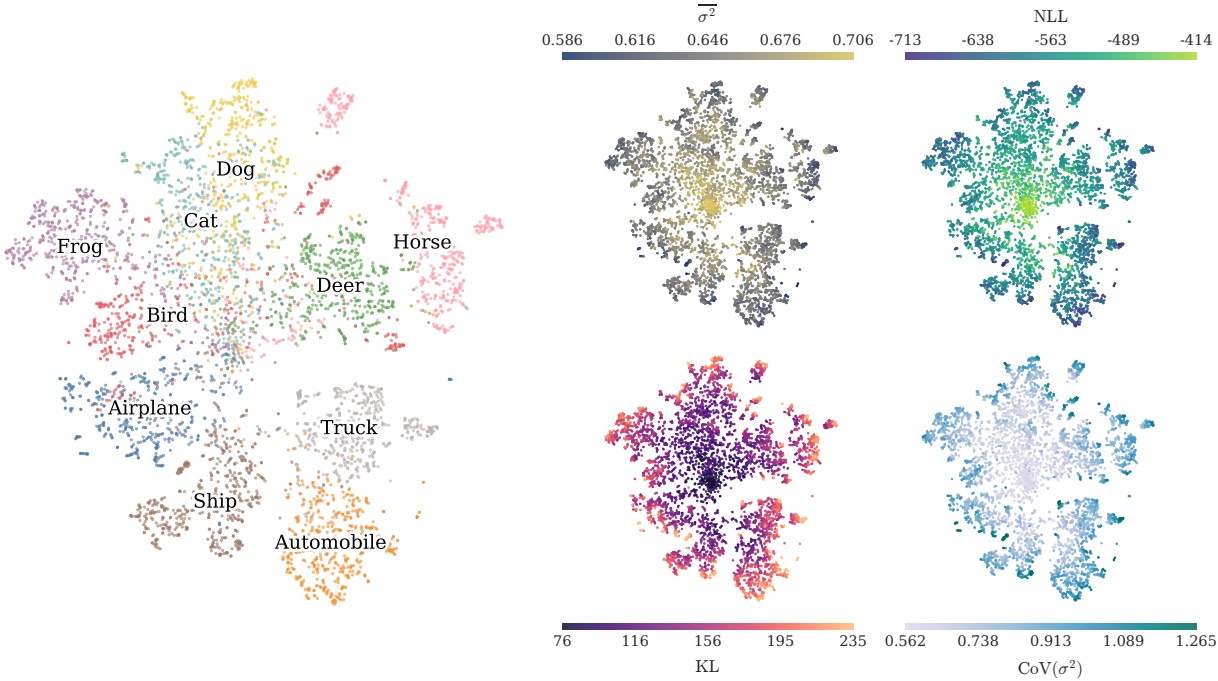

Figure 4: Geometry of posterior uncertainty on CIFAR–10. The left panel shows a t-SNE (van der Maaten and Hinton, 2008) of the learned representation colored by class. The four panels on the right visualize posterior-derived quantities over the same embedding geometry: the mean posterior variance $\overline{\sigma^2}$, the negative log-likelihood (NLL), the KL term, and the anisotropy of the posterior variance field, measured as the coefficient of variation (CoV) of the posterior variance.

Table 3: Posterior geometry as a function of class-center margin and within-class radius in normalized posterior-mean space on CIFAR–10. The upper section partitions samples into tertiles by margin, where low margin indicates weaker separation and high margin indicates cleaner separation. The lower section reports Spearman rank correlations ($\rho$) of each posterior statistic with margin and radius.

| | $\overline{\sigma^2}$ | **NLL** | **KL** | $\text{CoV}(\sigma^2)$ |
|---|---|---|---|---|
| *Margin tertiles* | | | | |
| Low | $0.6630 \pm 0.0237$ | $-531.55 \pm 57.52$ | $120.28 \pm 33.20$ | $0.6883 \pm 0.1450$ |
| Mid | $0.6436 \pm 0.0215$ | $-577.98 \pm 52.20$ | $153.85 \pm 38.02$ | $0.8203 \pm 0.1494$ |
| High | $0.6354 \pm 0.0209$ | $-610.68 \pm 54.64$ | $171.60 \pm 38.36$ | $0.8720 \pm 0.1433$ |
| *Spearman correlation ($\rho$)* | | | | |
| Margin | $-0.50$ | $-0.58$ | $0.58$ | $0.55$ |
| Radius | $-0.36$ | $-0.28$ | $0.54$ | $0.69$ |

whereas the more peripheral and visually separated regions exhibit larger deviation from it. Together with the patterns of variance and anisotropy, this suggests a transition from broader, more isotropic uncertainty near the centre of the embedding to smaller but more structured posterior uncertainty toward the periphery.

To quantify this effect directly in the original representation space, we compute two geometric measures in normalized posterior-mean space. For a sample with label $y_i$, the class-center margin is defined as:

$$m_i = \cos(\tilde{\mu}_i, c_{y_i}) - \max_{c \neq y_i} \cos(\tilde{\mu}_i, c), \tag{40}$$

where $\tilde{\mu}_i$ is the normalized posterior mean and $c_y$ is the centroid of class $y$ in the same space. Large margin indicates stronger separation from competing classes, while small margin indicates a more weakly separated region. The within-class radius is defined as $r_i = 1 - \cos(\tilde{\mu}_i, c_{y_i})$, measuring distance from the own-class centroid while ignoring competing classes.

All four posterior statistics vary monotonically with margin, confirming that samples with weaker class separation are associated with broader uncertainty and worse likelihood fit, but with lower KL and lower anisotropy, consistent with posteriors that remain closer to the prior. Conversely, well-separated samples have tighter overall uncertainty together with larger KL and more anisotropic posterior variance, indicating stronger deviation from the isotropic prior geometry. The posterior therefore does not respond to class geometry merely by inflating or shrinking a single variance scale. Its overall magnitude and its internal structure vary in opposite directions across the representation.

The correlation structure further differentiates the roles of these statistics. Anisotropy is most strongly associated with within-class radius ($\rho = 0.69$), whereas NLL is more strongly associated with margin than with radius ($-0.58$ versus $-0.28$). This suggests that NLL is more closely tied to inter-class separation, while anisotropy is more closely tied to position within the class cloud. Taken together, the evidence indicates that mean variance and NLL track uncertainty magnitude and fit relative to global separation, while KL and anisotropy capture a complementary aspect of posterior structure tied to the degree of departure from the prior. This separation between uncertainty scale and posterior structure is enabled by the variance tying between the posterior and the directional likelihood (Section 4.2), which allows the same per-feature scale $\boldsymbol{\sigma}^2$ to govern both the likelihood and the posterior.

## 5.4 Empirical assessment of probabilistic semantics

We next assess the probabilistic semantics induced by VJE in representation space by evaluating posterior-derived scores for out-of-distribution detection under the OpenOOD CIFAR–10 protocol, using CIFAR–100 and Tiny ImageNet as near-OOD datasets, and MNIST (LeCun et al., 1998), SVHN (Netzer et al., 2011), Textures (Cimpoi et al., 2014), and Places365 (Zhou et al., 2018) as far-OOD datasets (Yang et al., 2022; Zhang et al., 2024). No class labels are used at any stage of training or scoring. The primary score is the

Table 4: OOD detection on CIFAR–10 using posterior-derived scores ($\nu$=1.0, 200 epochs, three runs). NLL denotes the joint likelihood of Eq. (39); NLL (dir) uses the directional term only. Entries report AUROC (%) as mean $\pm$ std; **bold** indicates the column best and scores within one standard deviation of it. Summary **Avg** follows the OpenOOD convention and is computed as (Near + Far)/2.

| | Near-OOD | | Far-OOD | | | | Summary | | |
| Score | C-100 | TIN | MNIST | SVHN | Textures | Places | Near | Far | Avg |
|---|---|---|---|---|---|---|---|---|---|
| NLL | **88.1 ± 0.2** | **88.3 ± 0.3** | **97.4 ± 0.8** | **98.9 ± 0.2** | **96.7 ± 0.7** | **93.0 ± 0.3** | **88.2 ± 0.2** | **96.5 ± 0.2** | **92.4 ± 0.2** |
| NLL (dir) | **88.2 ± 0.2** | **88.5 ± 0.3** | **97.4 ± 0.9** | **98.9 ± 0.2** | **96.8 ± 0.6** | **93.1 ± 0.3** | **88.3 ± 0.2** | **96.6 ± 0.2** | **92.5 ± 0.2** |
| Tr($\sigma^2$) | 83.6 ± 0.4 | 85.7 ± 0.6 | 80.8 ± 12.2 | 91.3 ± 2.0 | 82.8 ± 2.0 | 89.0 ± 0.7 | 84.6 ± 0.5 | 86.0 ± 2.5 | 85.3 ± 1.2 |
| −KL | 84.8 ± 0.4 | 85.7 ± 0.3 | 70.5 ± 5.5 | 95.4 ± 0.6 | 76.4 ± 2.1 | 88.1 ± 0.5 | 85.3 ± 0.3 | 82.6 ± 0.9 | 84.0 ± 0.3 |
| −CoV($\sigma^2$) | 82.3 ± 0.5 | 81.8 ± 0.5 | 58.8 ± 9.2 | 93.6 ± 0.9 | 66.2 ± 2.2 | 82.1 ± 0.9 | 82.0 ± 0.4 | 75.2 ± 2.1 | 78.6 ± 1.2 |

NLL of the joint likelihood (Eq. (39)). Unless otherwise noted, all results in this subsection use $\nu$=1.0, 200 training epochs, and three runs.

Table 4 reports the NLL score alongside other posterior-derived statistics. The NLL score achieves $92.4 \pm 0.2$ OpenOOD average AUROC, with $88.2 \pm 0.2$ on near-OOD and $96.5 \pm 0.2$ on far-OOD. The directional NLL alone is nearly identical, indicating that the OOD signal is already largely present in the directional likelihood, while the radial term contributes little additional discrimination. The remaining uncertainty measures are all informative, though they all underperform the NLL score.

This is consistent with the geometric analysis in Section 5.3, where variance magnitude and anisotropy were shown to track different aspects of the posterior, neither of which alone captures the full structure

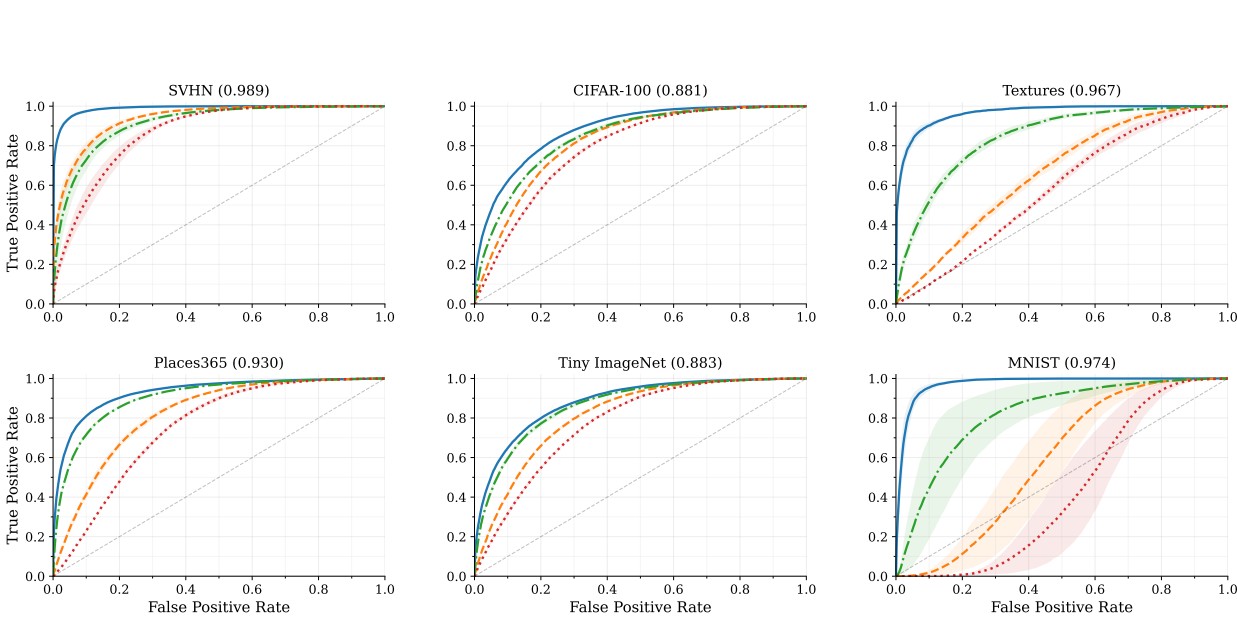

Figure 5: OOD detection ROC curves on CIFAR–10 for six OOD datasets. Each panel compares CIFAR–10 against one OOD dataset and reports the AUROC of the NLL score in the title. The plotted curves correspond to four posterior-derived scores: NLL, $-\text{CoV}(\sigma^2)$, $\text{Tr}(\sigma^2)$, and $-\text{KL}$. Shaded bands denote $\pm 1$ standard deviation across the three runs. Across all six datasets, the NLL score is the most consistent and reliable discriminator. This is especially evident for MNIST, where the alternative uncertainty measures exhibit markedly weaker or less stable separation, while NLL still yields near-perfect discrimination.

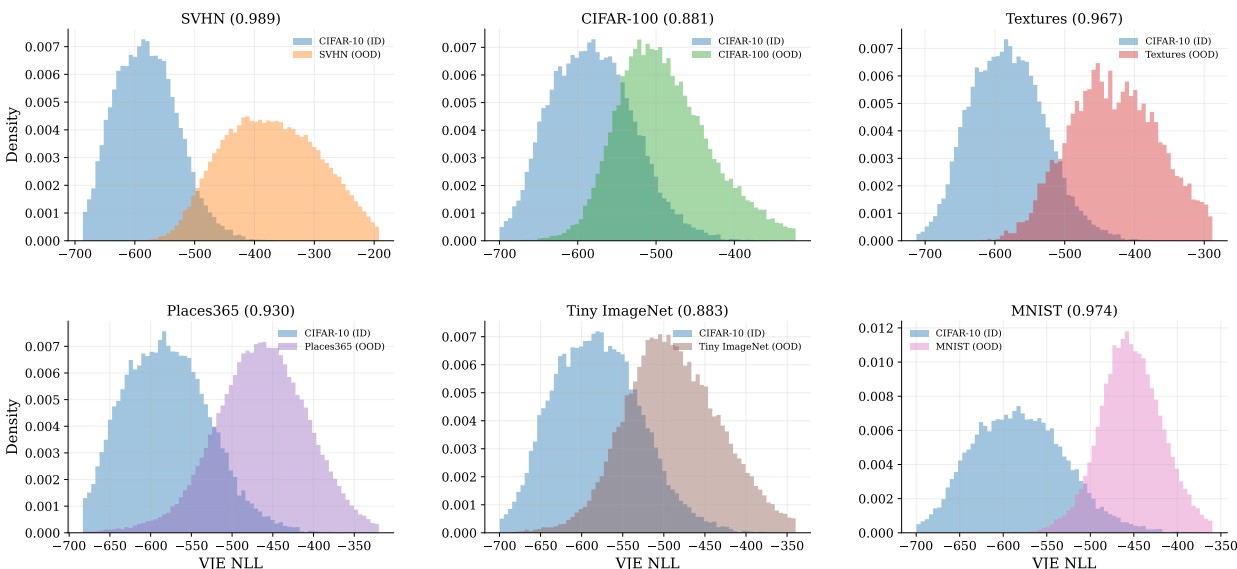

Figure 6: Distributions of the NLL score on CIFAR–10 and six OOD datasets. Each panel compares the CIFAR–10 in-distribution score distribution against one OOD dataset, with the corresponding AUROC shown in the title.

encoded by the likelihood. Notably, both KL and $\mathrm{CoV}(\sigma^2)$ require sign-flipping to be used as OOD scores, as OOD samples tend to have higher KL and lower anisotropy than in-distribution samples, indicating greater departure from the prior but with less structured variance. This is the same direction of effect observed in Section 5.3, where low-margin in-distribution samples were the ones closest to the prior anchor, and it extends naturally to OOD inputs that lie further from the learned class structure.

Figure 5 and the score distributions in Figure 6 show that the NLL-based signal is the most consistent across datasets. OOD samples shift toward larger NLL in all six cases, with especially clean separation for SVHN and MNIST and more overlap for the near-OOD datasets CIFAR–100 and Tiny ImageNet.

**Sensitivity to the likelihood tail parameter.** Table 5 shows that $\nu$=1.0 and $\nu$=3.0 perform essentially identically overall, and $\nu$=7.0 remains competitive. Performance degrades at $\nu$=20.0 and collapses at $\nu$=50.0, where the detector is near chance. As $\nu$ increases, the Student–$t$ likelihood approaches the Gaussian limit and the bounded-influence property discussed in Section 4 is lost: gradients no longer saturate for large residuals, and the representation fails to develop the uncertainty structure that makes the NLL score discriminative. At $\nu$=50.0, this corresponds to effective model collapse.

Table 5: NLL score AUROC (%) on CIFAR–10 OOD detection across the $\nu$-sweep ($\beta$=1.0, 200 epochs, three runs). Entries report AUROC (%) as mean $\pm$ std; **bold** indicates the column best and scores within one standard deviation of it. Summary **Avg** follows the OpenOOD convention and is computed as $(\mathrm{Near}+\mathrm{Far})/2$.

| | Near-OOD | | Far-OOD | | | | Summary | | |
|---|---|---|---|---|---|---|---|---|---|
| $\nu$ | C-100 | TIN | MNIST | SVHN | Textures | Places | Near | Far | Avg |
| 1 | **88.1 ± 0.2** | 88.3 ± 0.3 | **97.4 ± 0.8** | **98.9 ± 0.2** | **96.7 ± 0.7** | 93.0 ± 0.3 | 88.2 ± 0.2 | **96.5 ± 0.2** | **92.4 ± 0.2** |
| 3 | **88.3 ± 0.3** | **88.9 ± 0.2** | 96.8 ± 2.1 | 98.6 ± 0.2 | **96.7 ± 0.2** | 93.4 ± 0.1 | **88.6 ± 0.2** | 96.4 ± 0.4 | **92.5 ± 0.3** |
| 7 | 86.2 ± 0.6 | 87.7 ± 0.6 | **98.0 ± 0.7** | 98.0 ± 0.6 | **97.1 ± 0.3** | 93.0 ± 0.3 | 86.9 ± 0.6 | **96.5 ± 0.3** | 91.7 ± 0.2 |
| 20 | 86.0 ± 0.8 | 87.0 ± 0.8 | 96.9 ± 0.1 | 97.0 ± 0.4 | 95.6 ± 0.4 | 92.1 ± 0.7 | 86.5 ± 0.8 | 95.4 ± 0.4 | 91.0 ± 0.6 |
| 50 | 47.0 ± 4.6 | 49.7 ± 5.8 | 61.3 ± 26.1 | 44.2 ± 5.8 | 53.2 ± 7.2 | 50.3 ± 0.9 | 48.3 ± 5.0 | 52.3 ± 8.5 | 50.3 ± 6.5 |

Table 6: Reproduced VI-SimSiam OOD detection on CIFAR–10. Power-Spherical NLL is reported at 200, 400, and 800 epoch checkpoints; remaining scores are reported at 800 epochs. Entries report AUROC (%) as mean $\pm$ std over three runs. Summary **Avg** follows the OpenOOD convention and is computed as (Near + Far)/2.

| | Near-OOD | | Far-OOD | | | | Summary | | |
|---|---|---|---|---|---|---|---|---|---|
| Score | C-100 | TIN | MNIST | SVHN | Textures | Places | Near | Far | Avg |
| PS NLL (200) | $77.2 \pm 1.7$ | $83.8 \pm 1.5$ | $11.2 \pm 3.8$ | $92.4 \pm 0.5$ | $77.0 \pm 3.3$ | $85.9 \pm 0.4$ | $80.5 \pm 1.6$ | $66.6 \pm 0.2$ | $73.6 \pm 0.7$ |
| PS NLL (400) | $83.5 \pm 0.8$ | $88.7 \pm 1.1$ | $26.2 \pm 6.1$ | $95.0 \pm 0.7$ | $85.2 \pm 3.3$ | $89.4 \pm 1.1$ | $86.1 \pm 0.9$ | $74.0 \pm 2.7$ | $80.1 \pm 1.8$ |
| PS NLL (800) | $84.9 \pm 0.1$ | $89.8 \pm 0.6$ | $42.5 \pm 3.6$ | $97.6 \pm 0.4$ | $85.1 \pm 0.8$ | $90.1 \pm 0.3$ | $87.3 \pm 0.2$ | $78.8 \pm 0.7$ | $83.0 \pm 0.3$ |
| $-\kappa$ (800) | $84.9 \pm 0.1$ | $89.7 \pm 0.5$ | $42.3 \pm 2.3$ | $97.6 \pm 0.3$ | $84.5 \pm 1.0$ | $90.2 \pm 0.6$ | $87.3 \pm 0.3$ | $78.6 \pm 0.5$ | $83.0 \pm 0.2$ |
| PS ELBO (800) | $79.5 \pm 9.8$ | $86.4 \pm 5.4$ | $41.3 \pm 14.9$ | $95.5 \pm 4.1$ | $83.0 \pm 6.1$ | $85.2 \pm 8.7$ | $82.9 \pm 7.6$ | $76.2 \pm 8.4$ | $79.6 \pm 8.0$ |
| cos dist (800) | $61.8 \pm 17.0$ | $62.1 \pm 15.7$ | $72.5 \pm 31.4$ | $70.9 \pm 35.2$ | $73.1 \pm 20.2$ | $60.5 \pm 27.0$ | $61.9 \pm 16.3$ | $69.3 \pm 28.4$ | $65.6 \pm 22.4$ |

**Comparison to baselines.** We include a reproduced VI-SimSiam baseline (Nakamura et al., 2023) as the closest comparable non-contrastive probabilistic model. Table 6 shows that its Power-Spherical NLL improves substantially with training time, reaching $83.0 \pm 0.3$ OpenOOD average AUROC at 800 epochs. Its strongest detector is the likelihood-based score itself, with $-\kappa$ giving nearly identical results, while ELBO and especially cosine distance are weaker. This comparison also shows that, within the same VI-SimSiam framework, deterministic point-similarity is substantially less effective for OOD detection than the likelihood-based probabilistic scores, despite cosine distance being its central loss component. VI-SimSiam therefore also learns usable uncertainty structure under this evaluation, while VJE at 200 epochs (92.4) exceeds VI-SimSiam at 800 epochs (83.0) on the same benchmark.

Table 7: OpenOOD benchmark comparison on CIFAR–10. Near-OOD uses CIFAR–100 and Tiny ImageNet; Far-OOD uses MNIST, SVHN, Textures, and Places365. VJE is scored by the NLL of Eq. (39); VI-SimSiam by Power-Spherical NLL at 800 epochs. Generic baseline values are taken from Zhang et al. (2024); VI-SimSiam and VJE are our reproduced results. Avg follows the OpenOOD convention and is computed as the mean of the Near and Far aggregates.

| Method | Near | Far | Avg |
|---|---|---|---|
| MSP | 88.0 | 90.7 | 89.4 |
| ReAct | 87.1 | 90.4 | 88.8 |
| KNN | **90.6** | 93.0 | 91.8 |
| ViM | 88.7 | 93.5 | 91.1 |
| VI-SimSiam (reproduced) | $87.3 \pm 0.2$ | $78.8 \pm 0.7$ | $83.0 \pm 0.3$ |
| VJE (ours) | $88.2 \pm 0.2$ | $\mathbf{96.5 \pm 0.2}$ | $\mathbf{92.4 \pm 0.2}$ |

Table 7 places VJE in the broader context of the OpenOOD CIFAR–10 benchmark. The generic baseline values in that table are taken directly from OpenOOD v1.5 (Zhang et al., 2024), whereas the VI-SimSiam entry is our reproduced result. Under this comparison, VJE is competitive with strong generic baselines, achieving the highest far-OOD average while remaining within range of the best near-OOD methods. VI-SimSiam, by contrast, underperforms on far-OOD detection, which may reflect the limitations of an isotropic uncertainty mechanism that cannot specialize across representation dimensions.

Table 8 places VJE in the broader self-supervised OOD literature on the shared SVHN/CIFAR–100 benchmark, where prior baseline values are taken from Tack et al. (2020). Under this comparison, VJE remains competitive with the strongest reported baselines, trailing only CSI despite using no specialized pretext training and operating within a non-contrastive variational framework. These results indicate that VJE's probabilistic structure, while intrinsic to the model itself, remains competitive even against specialized self-supervised methods designed explicitly for this setting.

Table 8: Self-supervised OOD detection on the shared SVHN/CIFAR–100 benchmark, provided to situate VJE among other self-supervised representation learning approaches. The **Method family** column distinguishes specialized pretext-based methods, contrastive self-supervised learning, and variational self-supervised learning. All methods use ResNet–18 with their respective training regimes and do not use class labels. Prior baseline values are taken from Tack et al. (2020); VI-SimSiam and VJE are our reproduced results. Entries report AUROC (%).

| Method | *Method family* | SVHN | C-100 | Avg |
|---|---|---|---|---|
| Rot (Tack et al., 2020) | specialized pretext | $97.6 \pm 0.2$ | $79.0 \pm 0.1$ | 88.3 |
| Rot+Trans (Tack et al., 2020) | specialized pretext | $97.8 \pm 0.2$ | $82.3 \pm 0.2$ | 90.1 |
| GOAD (Tack et al., 2020) | specialized pretext | $96.3 \pm 0.2$ | $77.2 \pm 0.3$ | 86.8 |
| CSI (Tack et al., 2020) | contrastive SSL | $\mathbf{99.8 \pm 0.0}$ | $\mathbf{89.2 \pm 0.1}$ | **94.5** |
| VI-SimSiam (reproduced) | variational SSL | $97.6 \pm 0.4$ | $84.9 \pm 0.1$ | 91.2 |
| VJE (ours) | variational SSL | $98.9 \pm 0.2$ | $88.1 \pm 0.2$ | 93.5 |

## 5.5 Ablation studies

We perform ablations on the VJE objective along two axes: tail heaviness in the Student–$t$ likelihood, and the contribution of each loss component. All ablation experiments use 200 pretraining epochs with three seeds.

**Student–$t$ degrees of freedom.** To understand the role of tail heaviness, we sweep $\nu \in \{1.0, 3.0, 7.0, 20.0, 50.0, \infty\}$ across four datasets (Table 9).

Table 9: Ablation of Student–$t$ degrees of freedom $\nu$ across four datasets (200 epochs, $\beta$=1.0, three seeds). Entries report $k$–NN accuracy (%); **bold** indicates the column best and scores within one standard deviation of it. The Gaussian limit ($\nu \to \infty$) fails on all datasets; $\nu$=50.0 exhibits partial collapse with high variance across seeds.

| $\nu$ | CIFAR–10 | CIFAR–100 | STL–10 | Tiny ImageNet |
|---|---|---|---|---|
| 1 | $87.3 \pm 0.2$ | $\mathbf{56.3 \pm 0.2}$ | $\mathbf{80.9 \pm 0.1}$ | $\mathbf{34.2 \pm 0.4}$ |
| 3 | $87.5 \pm 0.2$ | $\mathbf{55.9 \pm 0.4}$ | $\mathbf{81.0 \pm 0.3}$ | $\mathbf{34.2 \pm 0.3}$ |
| 7 | $87.5 \pm 0.3$ | $\mathbf{55.8 \pm 0.6}$ | $80.4 \pm 0.2$ | $33.5 \pm 0.0$ |
| 20 | $\mathbf{88.0 \pm 0.3}$ | $55.1 \pm 0.2$ | $79.7 \pm 0.5$ | $31.9 \pm 0.3$ |
| 50 | $44.2 \pm 13$ | $7.5 \pm 1.9$ | $78.4 \pm 0.2$ | $20.9 \pm 16$ |
| $\infty$ | $16.3 \pm 11$ | $4.3 \pm 2.8$ | $13.8 \pm 4.1$ | $1.0 \pm 0.8$ |

For $\nu \in \{1.0, 3.0, 7.0, 20.0\}$, $k$–NN accuracy remains within a narrow band on all datasets, with $\nu$=1.0 and $\nu$=3.0 performing best overall. A sharp transition occurs between $\nu$=20.0 and $\nu$=50.0: on CIFAR–10, CIFAR–100, and Tiny ImageNet, $\nu$=50.0 exhibits high variance across seeds, indicating that some runs partially collapse while others survive. STL–10 is more resilient at $\nu$=50.0 (78.4%), likely because its larger image resolution provides a stronger augmentation signal. The Gaussian limit ($\nu \to \infty$) fails catastrophically on all datasets, confirming that the bounded influence of heavy-tailed likelihoods is essential for stable optimization. This is consistent with the OOD findings in Table 5, where the same transition produces near-chance detection at $\nu$=50.0. Based on these results, we adopt $\nu$=1.0 as the default throughout.

**Loss components.** To assess the contribution of the KL regularizer $\mathcal{L}_{\mathrm{KL}}$, directional term $\mathcal{L}_{\mathrm{dir}}$, and radial term $\mathcal{L}_{\mathrm{rad}}$, we ablate these components on three datasets with $\nu$=1.0. Table 10 reports $k$–NN accuracy and the dimension-averaged posterior variance $\overline{\sigma^2}$ for each configuration, with the full objective as reference.

The ablations reveal three qualitatively distinct regimes. The first regime corresponds to the full objective and the variant without the radial term ($\mathcal{L}_{\mathrm{dir}} + \mathcal{L}_{\mathrm{KL}}$). These are the only configurations that maintain strong

Table 10: Ablation of loss components on CIFAR–10, CIFAR–100, and STL–10 ($\nu$=1.0, 200 epochs, three seeds). Each configuration reports $k$–NN accuracy (%) and dimension-averaged posterior variance $\overline{\sigma^2}$. The full row reproduces the $\nu$=1.0 reference configuration from Table 9; ablated rows report $\overline{\sigma^2}$ as mean ± std over finite runs. STL–10 without KL contains one non-finite run; statistics are computed over the finite runs.

| | CIFAR–10 | | CIFAR–100 | | STL–10 | |
|---|---|---|---|---|---|---|
| **Objective** | $k$–NN | $\overline{\sigma^2}$ | $k$–NN | $\overline{\sigma^2}$ | $k$–NN | $\overline{\sigma^2}$ |
| $\mathcal{L}_{\mathrm{dir}} + \mathcal{L}_{\mathrm{rad}} + \mathcal{L}_{\mathrm{KL}}$ | 87.3±0.2 | 0.647 | 56.3±0.2 | 0.655 | 80.9±0.1 | 0.657 |
| $\mathcal{L}_{\mathrm{dir}} + \mathcal{L}_{\mathrm{KL}}$ | 87.4±0.3 | 0.648±0.000 | 56.0±0.4 | 0.655±0.003 | 80.7±0.3 | 0.657±0.001 |
| $\mathcal{L}_{\mathrm{dir}} + \mathcal{L}_{\mathrm{rad}}$ | 84.0±1.9 | 0.012±0.013 | 33.1±24.0 | 0.003±0.001 | 37.2±31.2 | 4.13±5.80 |
| $\mathcal{L}_{\mathrm{rad}} + \mathcal{L}_{\mathrm{KL}}$ | 31.8±2.0 | 0.987±0.004 | 8.7±1.0 | 0.992±0.001 | 26.3±0.7 | 1.003±0.008 |

discriminative performance and yield a nondegenerate posterior. The two are effectively indistinguishable: the $k$–NN differences are within 0.3 points and the posterior variance $\overline{\sigma^2}$ is essentially unchanged. The radial term therefore does not measurably contribute to representation quality; its role is geometric, tying the magnitude of latent samples to the encoder norm and preventing arbitrary rescaling.

The second regime arises when the KL regularizer is absent ($\mathcal{L}_{\mathrm{dir}} + \mathcal{L}_{\mathrm{rad}}$). Removing the KL yields unstable posterior behaviour and substantially weaker representation quality. On CIFAR–10, accuracy remains partially intact (84.0%) because the heavy-tailed directional likelihood at $\nu$=1.0 provides implicit regularization sufficient for a simpler task, though the posterior variance collapses toward zero ($\overline{\sigma^2}$=0.012). On CIFAR–100 and STL–10, some seeds maintain reasonable accuracy while others collapse completely, producing the large standard deviations of ±24 and ±31 points together with near-zero or highly unstable variance. This confirms that the KL term is essential for reliable training and for maintaining a nondegenerate posterior with structured uncertainty.

The third regime occurs when the directional likelihood is removed ($\mathcal{L}_{\mathrm{rad}} + \mathcal{L}_{\mathrm{KL}}$). With only the radial channel and prior regularization, the posterior converges to the isotropic prior ($\overline{\sigma^2} \approx 1$, effectively zero anisotropy) and accuracy drops to near-chance on CIFAR–100 and STL–10. Without the directional term, there is no mechanism to shape posterior anisotropy or to align latent representations with target directions. This demonstrates that the directional likelihood is the primary driver of both discriminative capacity and structured posterior uncertainty in VJE.

### 5.6 Synthesis

Across representation learning, geometry, out-of-distribution detection, and ablation experiments, we highlight three findings.

First, the variational objective does not compromise discriminative capacity. On ImageNet–1K, VJE achieves 68.2% top-1 linear accuracy (Table 1), placing it in the same range as strong deterministic baselines under the same epoch budget. On the smaller-scale benchmarks, VJE with EMA outperforms both reproduced deterministic baselines on all three datasets (Table 2).

Second, the learned posterior exhibits genuine geometric structure that goes beyond a scalar uncertainty measure. Section 5.3 shows that mean variance and NLL track inter-class separation, while KL and anisotropy track a complementary aspect of posterior structure tied to typicality relative to the prior. The NLL score of Eq. (39) integrates these effects and achieves 92.4 average AUROC on the OpenOOD benchmark (Table 4), outperforming all other posterior statistics.

Third, this structure depends on the interaction between the heavy-tailed likelihood and the diagonal posterior. The $\nu$-sweep (Table 9) shows that the model remains robust across $\nu \in \{1.0, 3.0, 7.0, 20.0\}$ but collapses as the likelihood approaches the Gaussian limit. The loss-component ablation (Table 10) shows that removing either the directional term or the KL regularizer leads to qualitative failure. Taken together with the representation-learning and OOD $\nu$-sweeps (Tables 9 and 5), these results suggest that while the model

remains robust across moderately heavy-tailed settings, $\nu = 1$ (the Cauchy case) provides the best general performance and overall trade-off between discriminative performance and probabilistic semantics.

Further supporting experiments are also provided in the appendices. Appendix C.1 empirically confirms the norm-direction coupling pathology that motivates the factorized likelihood, showing that the standard formulation undergoes posterior collapse on ImageNet–100. Appendix C.3 shows that replacing the diagonal posterior with an isotropic scalar-variance variant produces complete failure, with the posterior mean remaining at chance-level accuracy for all tested $\nu$, indicating that per-dimension variance is necessary for the model to learn structured uncertainty. Additionally, Appendix C.2 shows that a single Monte Carlo sample ($K{=}1$) is sufficient and that additional samples provide no measurable improvement, while Appendix C.4 shows that the computational overhead of the variational formulation is largely absorbed by the simpler inference network and the absence of auxiliary projection heads or comparable architectural components. To support reproducibility, Appendix A provides pseudocode for the forward pass and loss computation, together with additional implementation details.

## 6 Conclusion

In this work, we introduced **Variational Joint Embedding (VJE)**, a variational formulation of non-contrastive self-supervised learning. By using an amortized inference network and maximizing a symmetric conditional ELBO, VJE preserves the reconstruction-free training paradigm of joint embedding architectures while providing feature-wise uncertainty signals through an explicit variational posterior.

The framework rests on a likelihood that can be trained effectively in representation space, where naive formulations suffer from norm-induced pathologies that couple angular alignment with embedding magnitude. We addressed this by factorizing the likelihood into decoupled directional and radial components, reparameterizing the radial term as a norm difference, and anchoring the geometry through analytic KL regularization. This construction removes the coupling by design, as confirmed by direct comparison against an unfactorized alternative (Appendix C.1). Empirically, VJE remains competitive with standard non-contrastive baselines on representation learning benchmarks, while producing non-degenerate posteriors whose likelihood semantics yield strong performance on out-of-distribution detection (Section 5.4) without task-specific modifications.

Our theoretical analysis in Appendix B further clarifies the relationship between this likelihood-based formulation and standard pointwise objectives used in non-contrastive learning. In particular, it distinguishes the normalized conditional modelling primitive underlying VJE from the pointwise compatibility primitive of JEPA-style objectives (LeCun, 2022), and shows how common discrepancy-based objectives can be recovered only under explicit restrictions at the objective level.

**Limitations and future directions.** The uncertainty developed in this work is intrinsic to representation space, characterizing how confidently the model accounts for an input within its learned geometry. Projecting this uncertainty into downstream task spaces, for example to obtain calibrated predictive distributions, is a separate modelling problem beyond the scope of the present work, but a natural next step.

Although our experiments focus on augmented views in vision, the underlying formulation requires only paired observations of a shared latent signal, and can be extended to multimodal settings and other input domains. A related direction is adapting the framework to patch-based or hierarchical architectures, particularly Vision Transformers (Dosovitskiy et al., 2021), where the likelihood and inference network currently defined on a single global embedding would need to operate over collections of token-level representations.

Finally, while we instantiate the likelihood using heavy-tailed Student–$t$ factors, the framework is not restricted to this family. Exploring alternative likelihood kernels may further refine the robustness and geometric expressiveness of the model across domains and modalities.

## Acknowledgements

We thank the anonymous reviewers and our action editor, Ole Winther, for their careful review and valuable feedback, which led to many improvements in this work.

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

## A  Pseudocode and implementation details

This appendix provides pseudocode for the VJE objective described in Sections 3 and 4, followed by implementation details to assist with reproducibility. The routines implement the symmetric conditional ELBO of Eq. (38) using the factorized directional and radial Student–$t$ likelihoods together with a diagonal Gaussian variational posterior. All likelihood terms are evaluated with the target view detached (stop-gradient or EMA), and constants independent of $(\boldsymbol{\mu}, \boldsymbol{\sigma}^2)$ are omitted.

**Directional negative log-likelihood.**  This routine evaluates the directional Student–$t$ negative log-likelihood on the unit sphere, corresponding to Eq. (27). It computes the geodesic log-map, the Schur-complement Mahalanobis distance, the tangent-space log-determinant, and the exponential-map Jacobian correction.

---

**Algorithm 1** Directional negative log-likelihood (Student–$t$ on $\mathbb{S}^{D-1}$)

---

**Require:** Target embedding $\mathbf{z}$, sample $\mathbf{s}$, variance $\boldsymbol{\sigma}^2$, degrees of freedom $\nu$, dimension $D$

1: $\hat{\mathbf{z}} \leftarrow \mathrm{normalize}(\mathbf{z});\quad \hat{\mathbf{s}} \leftarrow \mathrm{normalize}(\mathbf{s});\quad \mathbf{n} \leftarrow \hat{\mathbf{s}}$          ▷ *Unit directions and normal*
2: $\cos\theta \leftarrow \hat{\mathbf{z}}^\top \hat{\mathbf{s}};\quad \theta \leftarrow \arccos(\cos\theta);\quad \sin\theta \leftarrow \sqrt{1 - \cos^2\theta}$       ▷ *Geodesic distance*
3: $\mathbf{t} \leftarrow (\theta/\sin\theta)\,(\hat{\mathbf{z}} - \cos\theta\,\hat{\mathbf{s}})$          ▷ *Log-map: $\mathbf{t} \in T_{\hat{\mathbf{s}}}\mathbb{S}^{D-1}$*
4: $\mathbf{w} \leftarrow \boldsymbol{\sigma}^{-2}$          ▷ *Precision weights*
5: $c \leftarrow \sum_d \hat{s}_d^2\, w_d$          ▷ $\mathbf{n}^\top \Sigma^{-1} \mathbf{n}$
6: $a \leftarrow \sum_d t_d^2\, w_d;\quad b \leftarrow \sum_d t_d\, \hat{s}_d\, w_d$
7: $Q \leftarrow a - b^2/c$          ▷ *Schur-complement Mahalanobis (Eq. 21)*
8: $\mathrm{logdet} \leftarrow \frac{1}{2}\sum_d \log \sigma_d^2 + \frac{1}{2}\log c$       ▷ *Half tangent-space log-determinant (Eq. 22)*
9: $\mathrm{jac} \leftarrow (D-2)(\log\sin\theta - \log\theta)$          ▷ *Exp-map Jacobian (Eq. 25)*
10: $k \leftarrow D - 1$
11: **return** $\frac{1}{2}(\nu + k)\log\!\big(1 + Q/\nu\big) + \mathrm{logdet} + \mathrm{jac}$

---

**Radial negative log-likelihood.**  This routine evaluates the radial Student–$t$ negative log-likelihood on the norm difference $\Delta r = \|\mathbf{z}\| - \|\mathbf{s}\|$ as introduced in Section 4.3.

---

**Algorithm 2** Radial negative log-likelihood (Student–$t$ on $\Delta r$)

---

**Require:** Target embedding $\mathbf{z}$, sample $\mathbf{s}$, degrees of freedom $\nu$

1: $r_z \leftarrow \|\mathbf{z}\|;\quad r_s \leftarrow \|\mathbf{s}\|$
2: $\Delta r \leftarrow r_z - r_s$
3: **return** $\frac{1}{2}(\nu + 1)\log\!\big(1 + (\Delta r)^2/\nu\big)$

---

**KL divergence.**  This routine computes the analytic KL divergence between a diagonal Gaussian posterior and the standard Gaussian prior, as in Eq. (7).

---

**Algorithm 3** KL divergence (diagonal Gaussian vs. standard Normal)

---

**Require:** Mean vector $\boldsymbol{\mu}$, variance vector $\boldsymbol{\sigma}^2$

1: **return** $\frac{1}{2}\sum_k \big(\sigma_k^2 + \mu_k^2 - 1 - \log\sigma_k^2\big)$

---

**VJE training step.**  This routine performs one symmetric VJE update, combining the directional and radial likelihood terms with the KL regularizer as in Eq. (8).

---

**Algorithm 4** One training step of VJE

---

**Require:** Encoder $f_\theta$, inference network $g_\phi$, views $(x_1, x_2)$; weights $\beta, \nu$; number of MC samples $K$

1: $\mathbf{z}_1 \leftarrow f_\theta(x_1); \quad \mathbf{z}_2 \leftarrow f_\theta(x_2)$
2: $(\boldsymbol{\mu}_1, \boldsymbol{\sigma}_1^2) \leftarrow g_\phi(\mathbf{z}_1); \quad (\boldsymbol{\mu}_2, \boldsymbol{\sigma}_2^2) \leftarrow g_\phi(\mathbf{z}_2)$
3: **for** $k = 1, \ldots, K$ **do**
4: $\quad \boldsymbol{\epsilon}_i \sim \mathcal{N}(\mathbf{0}, \mathbf{I}); \quad \mathbf{s}_i^{(k)} \leftarrow \boldsymbol{\mu}_i + \sqrt{\boldsymbol{\sigma}_i^2} \odot \boldsymbol{\epsilon}_i \quad$ for $i = 1, 2$
5: $\ell_{\text{dir}} \leftarrow \frac{1}{2K} \sum_{k=1}^{K} \big[ \text{nll\_dir}(\mathbf{z}_2.\text{detach}(), \mathbf{s}_1^{(k)}, \boldsymbol{\sigma}_1^2, \nu, D)$
$\quad\quad\quad\quad\quad\quad\quad\quad + \text{nll\_dir}(\mathbf{z}_1.\text{detach}(), \mathbf{s}_2^{(k)}, \boldsymbol{\sigma}_2^2, \nu, D) \big]$
6: $\ell_{\text{rad}} \leftarrow \frac{1}{2K} \sum_{k=1}^{K} \big[ \text{nll\_rad}(\mathbf{z}_2.\text{detach}(), \mathbf{s}_1^{(k)}, \nu)$
$\quad\quad\quad\quad\quad\quad\quad\quad + \text{nll\_rad}(\mathbf{z}_1.\text{detach}(), \mathbf{s}_2^{(k)}, \nu) \big]$
7: $\ell_{\text{KL}} \leftarrow \frac{1}{2} \big[ \text{kld}(\boldsymbol{\mu}_1, \boldsymbol{\sigma}_1^2) + \text{kld}(\boldsymbol{\mu}_2, \boldsymbol{\sigma}_2^2) \big]$
8: **return** $\ell_{\text{dir}} + \ell_{\text{rad}} + \beta \, \ell_{\text{KL}}$

---

## A.1 Inference network architecture

The inference network $g_\phi$ described in Section 3 and used in experiments of Section 5 is implemented as a two-layer bottleneck MLP followed by two separate linear output heads. Concretely, for an encoder with output dimension $D$ and bottleneck width ratio $r$ (yielding hidden dimension $H = \lfloor rD \rfloor$), the architecture is:

$$\mathbf{z} \xrightarrow{\text{Linear}(D,H)} \text{LN} \rightarrow \text{ReLU} \xrightarrow{\text{Linear}(H,H)} \text{LN} \rightarrow \text{ReLU} \rightarrow \begin{cases} \text{Linear}(H, D) & \rightarrow \boldsymbol{\mu} \\ \text{Linear}(H, D_v) \rightarrow \text{Softplus} & \rightarrow \boldsymbol{\sigma}^2 \end{cases}$$

where LN denotes Layer Normalization (Ba et al., 2016) and $D_v = D$ for the diagonal (anisotropic) parameterization. All linear layers in the bottleneck use `bias=False`, since the subsequent Layer Normalization absorbs any bias. The output heads retain bias terms. We use a bottleneck width ratio of $r=0.25$, yielding $H=128$ for ResNet–18 ($D=512$) and $H=512$ for ResNet–50 ($D=2048$).

The Softplus activation $\boldsymbol{\sigma}^2 = \log(1 + \exp(\cdot))$ ensures strict positivity of the variance, supplemented by a small additive floor $\varepsilon = 10^{-5}$. The bottleneck and output heads are initialized with Xavier uniform (Glorot and Bengio, 2010), while the ResNet (He et al., 2016) encoder retains its default random initialization (He et al., 2015).

## A.2 Additional implementation notes

- **ELBO structure.** Likelihood terms use $\mathbf{z}_j.\text{detach}()$, implementing the conditional ELBO of Section 4.4 and Eq. (38), where each view predicts the latent representation of the opposite view under fixed-observation semantics. The same semantics can equivalently be implemented via an EMA target encoder in place of stop-gradient.

- **Numerical stability.** A small constant (e.g., $10^{-6}$) is used inside `normalize` (via $\hat{\mathbf{z}} = \mathbf{z}/\max(\|\mathbf{z}\|, \varepsilon)$) and as a floor on $\boldsymbol{\sigma}^2$ to avoid division by zero.

- **Radial scale.** The radial likelihood uses $\Delta r = \|\mathbf{z}\| - \|\mathbf{s}\|$ with the scale fixed to $\lambda = 1$, as described in Section 4.3.

- **Monte Carlo sampling.** Each expectation under $q(\mathbf{s} \mid \mathbf{z})$ in the likelihood term can be approximated with $K$ reparameterized samples per view, averaged over the $K$ draws. In all reported experiments we use $K=1$, as we note in Appendix C.2 that increasing $K$ yields no measurable improvement.

- **Log-variance centering.** For high-dimensional embeddings ($D=2048$), the log-determinant $\frac{1}{2} \sum_d \log \sigma_d^2$ in Eq. (27) grows as $O(D)$ and can potentially lead to numerical destabilization in small variance regimes. To avoid this, the variance can be normalized to unit geometric mean within the directional likelihood, $\sigma_{c,d}^2 = \sigma_d^2/(\prod_j \sigma_j^2)^{1/D}$, so that $\sum_d \log \sigma_{c,d}^2 = 0$ by construction. This reduces the log-determinant contribution to $O(1)$ while leaving relative anisotropy and probabilistic

semantics unchanged. The KL divergence (Eq. (7)) continues to use the original $\boldsymbol{\sigma}^2$, so absolute scale is still regularized. In practice, this centering had no measurable impact in our setting and serves only as a numerical stabilizer. We apply this only for the ImageNet–1K evaluation (ResNet–50 $D=2048$) experiment, while all other experiments are trained stably without it as they use ResNet-18 ($D=512$).

## B Energy-based Predictive Learning and Likelihood-based Modelling

### B.1 Overview and scope

Predictive joint embedding in non-contrastive self-supervised learning admits two distinct formulations. The first is an *energy-based* or compatibility formulation (LeCun, 2022), in which training minimizes a pointwise discrepancy between a prediction and a target embedding, without specifying an explicit normalized density over embeddings. The second is the *likelihood-based* formulation that we adopt in Variational Joint Embedding (VJE), in which a conditional likelihood is specified in representation space and optimized through a conditional evidence lower bound (ELBO) (Section 4). While these paradigms may share architectural motifs (e.g., asymmetric branches, auxiliary heads, and momentum/target encoders (Grill et al., 2020; He et al., 2020; Chen et al., 2020b)), they optimize different primitives and accordingly assign different semantic roles to their components.

In this appendix, we make an objective-level comparison between the primitive used by energy-based (deterministic) non-contrastive objectives and our VJE formulation. Architectural patterns vary across methods, and we do not conflate them with the energy-based or likelihood-based formulation itself, except where the likelihood-based formulation assigns a specific semantic role to the inference network and the target branch.

Following the notation defined in Section 3, given two related views $x_1, x_2$ of an input $x$, we denote the encoder embedding for view $i$ by $\mathbf{z}_i = f_\theta(x_i)$, the variational posterior by $q_i(\mathbf{s} \mid \mathbf{z}_i)$ with parameters $(\boldsymbol{\mu}_i, \boldsymbol{\sigma}_i^2)$ produced by the inference network $g_\phi$, the representation-space observation by $y_j = (\hat{\mathbf{z}}_j, \|\mathbf{z}_j\|)$, and the radial residual by $\Delta r_{ij} = \|\mathbf{z}_j\| - \|\mathbf{s}_i\|$ as in the factorized likelihood of Eq. (35). Scoring is performed symmetrically over $(i, j) \in \{(1, 2), (2, 1)\}$.

### B.2 Energy-based objectives and variational likelihoods

A typical joint-embedding objective takes the form of a pointwise discrepancy between a prediction and a target embedding:

$$\mathcal{L}_{\text{JEPA}}(x_i, x_j) = d\big(g_\phi(\mathbf{z}_i), \mathbf{z}_j\big). \tag{41}$$

Here $d(\cdot, \cdot)$ is commonly a squared-error or cosine-style discrepancy. Optimization proceeds directly on this pointwise score, shaping embedding geometry by enforcing low discrepancy for compatible pairs. While such objectives can be related formally to energies and unnormalized conditional models, they are typically optimized as pointwise compatibility losses on embedding pairs rather than as likelihoods.

To develop VJE, we instead take a conditional subdensity as the primitive. We specify $\tilde{p}_\psi(y_j \mid \mathbf{s}_i, \boldsymbol{\sigma}_i^2)$ on the representation-space observation $y_j$, and we optimize this model through a symmetric conditional ELBO by marginalizing latent codes to obtain the objective:

$$\overline{\mathcal{F}}^{(\beta)} = \frac{1}{2} \sum_{(i,j)\in\{(1,2),(2,1)\}} \left\{ \mathbb{E}_{\mathbf{s}_i \sim q_i} \left[ \log \tilde{p}_\psi(y_j \mid \mathbf{s}_i, \boldsymbol{\sigma}_i^2) \right] - \beta \, \mathcal{L}_{\text{KL}}(q_i \| p) \right\}, \tag{42}$$

where training minimizes $-\overline{\mathcal{F}}^{(\beta)}$ (Eq. (8)). The directional factor is derived from a normalized spherical reference construction on $\mathbb{S}^{D-1}$, while training evaluates the directional subdensity developed in Section 4.2. The negative log-likelihood terms therefore have explicit log-density semantics within the associated model, without introducing a partition function over embeddings.

### B.3 Representation semantics: inference and fixed observations

A likelihood-based objective assigns specific semantic roles to components that otherwise appear as training heuristics in energy-based objectives.

**Stop-gradient and fixed-observation semantics.** In VJE, the likelihood term evaluates the log-density of a target-side observation $y_j = (\hat{\mathbf{z}}_j, \|\mathbf{z}_j\|)$, while $\mathbf{z}_j$ is produced by a trainable encoder. In probabilistic modelling, observations are treated as fixed while model parameters are optimized to explain them. In representation learning, $y_j$ changes across training as the encoder evolves, and we enforce the intended semantics by treating $y_j$ as fixed within each update step for the likelihood term. This can be implemented either by stop-gradient on the target branch or equivalently by a target encoder held fixed during the update, including EMA variants; both realize the same fixed-observation conditioning required by the conditional ELBO.

To make this explicit, we differentiate the one-directional negative ELBO likelihood term and omit the KL term for clarity:

$$\mathcal{L}_{i \to j}^{\mathrm{NLL}}(\theta, \phi) = \mathbb{E}_{\mathbf{s}_i \sim q_i(\cdot \mid \mathbf{z}_i(\theta))} \left[ -\log \tilde{p}_\psi \big( y_j(\theta) \mid \mathbf{s}_i, \boldsymbol{\sigma}_i^2 \big) \right], \qquad y_j(\theta) = (\hat{\mathbf{z}}_j(\theta), \|\mathbf{z}_j(\theta)\|). \tag{43}$$

Taking gradients with respect to $\theta$ yields two pathways:

$$\nabla_\theta \mathcal{L}_{i \to j}^{\mathrm{NLL}} = \mathbb{E}_{\mathbf{s}_i \sim q_i} \left[ \nabla_{\mathbf{s}_i} \left( -\log \tilde{p}_\psi \big( y_j \mid \mathbf{s}_i, \boldsymbol{\sigma}_i^2 \big) \right) \nabla_\theta \mathbf{s}_i \right] + \mathbb{E}_{\mathbf{s}_i \sim q_i} \left[ \nabla_{y_j} \left( -\log \tilde{p}_\psi \big( y_j \mid \mathbf{s}_i, \boldsymbol{\sigma}_i^2 \big) \right) \nabla_\theta y_j \right]. \tag{44}$$

Our intended semantics correspond to retaining only the first pathway, in which model parameters are updated to explain a fixed observation under the conditional likelihood. Detaching the target branch in Eq. (8) sets $\nabla_\theta y_j = \mathbf{0}$ for the likelihood term. Allowing gradients to flow through both pathways changes the semantics of the objective by introducing a route in which the scored observation is itself modified by the same likelihood term.

**Amortized inference network.** Optimizing an ELBO requires an explicit variational posterior $q(\mathbf{s} \mid \mathbf{z})$ (Kingma and Welling, 2014; Rezende et al., 2014). In VJE, the inference network $g_\phi$ parameterizes this posterior by producing $(\boldsymbol{\mu}_i, \boldsymbol{\sigma}_i^2)$, enabling reparameterized sampling and efficient optimization of the ELBO. Deterministic objectives in energy-based joint embedding can be interpreted as learning point estimates, while VJE maintains an explicit distributional posterior that supports marginalization and density-based scoring.

**Objective and architecture.** The comparison above is objective-level, contrasting a pointwise energy score with a conditional subdensity model. Many non-contrastive baselines apply their loss in a projected space. We omit a separate projection head since the likelihood semantics in VJE are attached directly to the geometry of the encoder embedding space, and both the posterior $q_i(\mathbf{s} \mid \mathbf{z}_i)$ and the likelihood $\tilde{p}_\psi(y_j \mid \mathbf{s}_i, \boldsymbol{\sigma}_i^2)$ are defined in relation to $\mathbf{z}$ (Section 3).

### B.4 Unifying specific objectives as boundary configurations

With the primitives separated, we show two objective-level correspondences that recover common pointwise losses as boundary configurations under explicit likelihood choices and limits. These correspondences concern objective functions rather than full training pipelines, and they are not intended to claim complete equivalence of optimization dynamics.

**Energy-to-likelihood mapping and partition functions.** Given an energy $E(\mathbf{z}_j; \mathbf{z}_i)$, one may define an unnormalized conditional model $\tilde{p}(\mathbf{z}_j \mid \mathbf{z}_i) \propto \exp(-E(\mathbf{z}_j; \mathbf{z}_i))$. The corresponding normalized conditional density is:

$$p(\mathbf{z}_j \mid \mathbf{z}_i) = \frac{\exp(-E(\mathbf{z}_j; \mathbf{z}_i))}{Z(\mathbf{z}_i)}, \qquad Z(\mathbf{z}_i) = \int \exp(-E(\mathbf{z}; \mathbf{z}_i)) \, d\mathbf{z}. \tag{45}$$

Its negative log-likelihood decomposes as:

$$-\log p(\mathbf{z}_j \mid \mathbf{z}_i) = E(\mathbf{z}_j; \mathbf{z}_i) + \log Z(\mathbf{z}_i). \tag{46}$$

This identity explains why pointwise energy minimization does not generally coincide with likelihood maximization unless $\log Z(\mathbf{z}_i)$ is handled, for example by exact computation or sampling-based approximation, consistent with the energy-based predictive viewpoint of LeCun (2022).

**Recovering squared-error objectives.** A squared-error JEPA objective arises from our formulation when we replace the representation-space likelihood by an isotropic Gaussian density on embeddings, $p^{\mathcal{N}}(\mathbf{z}_j \mid \mathbf{s}_i) = \mathcal{N}(\mathbf{z}_j; \mathbf{s}_i, \lambda I)$, keep $\lambda > 0$ fixed, score the observation $\mathbf{z}_j$ rather than $y_j = (\hat{\mathbf{z}}_j, \|\mathbf{z}_j\|)$, take the point-estimate posterior limit $\boldsymbol{\sigma}_i^2 \to \mathbf{0}$ so that $\mathbf{s}_i \to \boldsymbol{\mu}_i$, and remove KL regularization by setting $\beta \to 0$. Under these limits and up to additive constants, the one-directional objective reduces to:

$$\mathcal{L}_{i \to j} \ \to \ \frac{1}{2\lambda} \|\mathbf{z}_j - \boldsymbol{\mu}_i\|^2, \qquad \boldsymbol{\mu}_i = g_\phi(\mathbf{z}_i), \tag{47}$$

which is a squared-error compatibility loss. Averaging symmetrically over both directions recovers the standard squared-error JEPA objective, and the same reduction applies when $\mathbf{z}_j$ represents a collection of patch embeddings, yielding the corresponding I-JEPA loss (Assran et al., 2023). The fixed-scale restriction is essential, since removing the KL term while retaining a learned per-sample scale no longer coincides with a deterministic squared-error energy, a distinction that is consistent with the variance degeneracy observed in the "without KL" ablation in Section 5.5.

**Recovering directional objectives.** A cosine-style objective arises from the directional channel when we restrict to the directional NLL of Eq. (27), fix the scale to be isotropic ($\boldsymbol{\sigma}_i^2 \equiv \mathbf{1}$), take the Gaussian limit $\nu \to \infty$ (Eq. (11)), take the point-estimate posterior limit, and remove the KL and radial terms. Under isotropic variance, the Schur-complement Mahalanobis distance reduces to the squared geodesic distance $\theta^2$ and the tangent-space log-determinant is constant. For small $\theta$, and treating the exp-map Jacobian as a fixed curvature correction, the directional NLL simplifies to:

$$\ell_{\mathrm{dir}} \ \to \ \frac{1}{2} \|\hat{\boldsymbol{\mu}}_i - \hat{\mathbf{z}}_j\|^2 = 1 - \hat{\boldsymbol{\mu}}_i^\top \hat{\mathbf{z}}_j, \tag{48}$$

which is equivalent to cosine alignment. Summed symmetrically over both directions, this reproduces a SimSiam-style objective at the level of the loss (Chen and He, 2021). SimSiam's empirical behaviour, however, also depends on its architectural asymmetry and optimization dynamics.

## B.5 Geometric regularization

The formulations also differ in how embedding geometry is regulated. Energy-based objectives are commonly paired with architectural stabilization, such as stop-gradient or momentum targets, and may additionally include explicit penalties on batch statistics. VICReg and Barlow Twins, for example, enforce representation diversity through variance or covariance constraints computed over batches (Bardes et al., 2022; Zbontar et al., 2021). These mechanisms constrain aggregate embedding geometry but do not specify a conditional likelihood on representation-space observations.

In VJE, geometric regularization follows from the probabilistic structure. The analytic KL term in Eq. (8) anchors each instance-conditioned posterior to the explicit isotropic Gaussian prior $p(\mathbf{s}) = \mathcal{N}(\mathbf{0}, I)$ (Eq. (7)). Moreover, we share the posterior scale $\boldsymbol{\sigma}_i^2$ with the directional likelihood scale (Section 4.2), so the same feature-wise parameters govern both sampling in $q_i(\mathbf{s})$ and weighting in the directional likelihood. This yields a prior-relative notion of geometry within a likelihood-based model and underlies scoring mechanisms such as the joint likelihood score used for out-of-distribution detection in Section 5.4.

LeJEPA (Balestriero and LeCun, 2025) provides a complementary perspective by advocating an isotropic Gaussian target geometry for embeddings and introducing SIGReg as an explicit global regularizer that encourages this structure. The relationship to VJE is geometric rather than a strict reduction, since LeJEPA enforces isotropy through a deterministic regularizer on encoder outputs, whereas VJE encourages prior-relative structure through an instance-conditioned posterior and analytic KL within a likelihood-based model.

## C Supplementary experiments and diagnostics

### C.1 Empirical analysis of standard and factorized likelihoods

To examine the norm-induced pathologies that motivate the polar decomposition in Section 4, we compare two likelihood formulations within the same VJE framework. The *factorized* formulation used throughout the paper decomposes the conditional likelihood into independent directional and radial terms, whereas the *standard* formulation evaluates a multivariate Student–$t$ density directly on the embedding vector, coupling direction and magnitude. All other settings are held fixed ($\nu$=1.0, $\beta$=1.0). We launch three runs per setting. CIFAR–10, CIFAR–100, and STL–10 are trained with ResNet–18 for 200 epochs, and ImageNet-100 with ResNet–50 for 100 epochs. Table 11 summarizes the end-of-training behaviour across all four datasets.

Table 11: End-of-training diagnostics for factorized and standard Student–$t$ likelihoods ($\nu$=1.0, $\beta$=1.0). ImageNet-100 uses ResNet–50 ($D$=2048); all others use ResNet–18 ($D$=512). We launch three runs per setting; under the standard likelihood, 2/3 ImageNet-100 runs and 1/3 STL–10 runs diverge to NaN, so reported values are averaged over successful runs only. $\overline{\sigma^2}$ denotes the dimension-averaged posterior variance.

| | Factorized | | | Standard | | |
|---|---|---|---|---|---|---|
| **Dataset** | $k$–**NN** | $\overline{\sigma^2}$ | **KLD** | $k$–**NN** | $\overline{\sigma^2}$ | **KLD** |
| ImageNet-100 | 71.0 | 0.630 | 595 | 6.3 | 0.110 | 1655 |
| CIFAR–10 | 87.3 | 0.647 | 141 | 86.8 | 0.245 | 286 |
| CIFAR–100 | 56.3 | 0.653 | 128 | 55.7 | 0.268 | 248 |
| STL–10 | 80.9 | 0.657 | 142 | 81.2 | 0.414 | 191 |

The pathology is most severe at larger scale. On ImageNet-100, the standard formulation undergoes catastrophic posterior collapse relative to the factorized variant (Table 11, Figure 7). Its average variance contracts from 0.33 to 0.11, the KL rises from 712 to 1655, and the effective rank (Roy and Vetterli, 2007) of the representation drops to 3 out of 2048 dimensions. Over the same training run, $k$–NN accuracy on the encoder output $z$ declines from 8.7% to 6.3%, while $k$–NN on the posterior mean $\boldsymbol{\mu}$ remains at chance level throughout. Under identical conditions, the factorized model trains normally, reaches 71.0% $k$–NN accuracy, maintains $\bar{\sigma}^2 \approx 0.63$, and reaches an effective rank of 991.

On the smaller benchmarks, the same mechanism is present but does not yet develop into full collapse. Table 11 shows a consistent pattern on CIFAR–10 and CIFAR–100, where the standard formulation yields

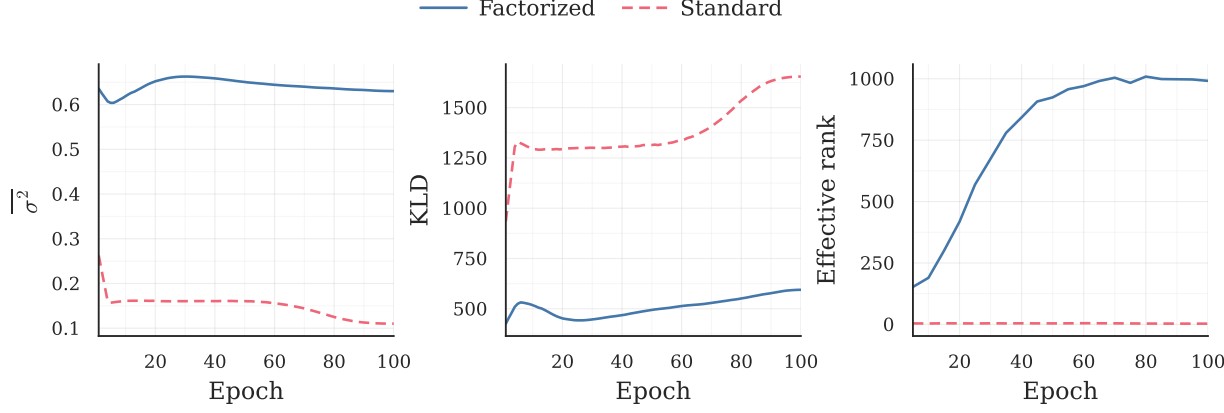

Figure 7: Training dynamics on ImageNet-100 (ResNet–50, 100 epochs) under the factorized (solid) and standard (dashed) Student–$t$ likelihood. The standard formulation progressively suppresses posterior variance, inflates the KL, and collapses the representation to a tiny effective subspace.

substantially lower posterior variance and higher KL, together with modestly worse $k$–NN accuracy. STL–10 is more mixed at the endpoint, but the standard formulation still exhibits lower variance, higher KL, and one divergence out of three runs. Taken together, these smaller-scale results suggest that the pathology is already active while its effects become catastrophic at larger scales.

To probe this mechanism more directly, Figure 8 evaluates both trained models under the same directional–radial decomposition. For the factorized model these terms are part of the optimized objective, while for the standard model they are post hoc quantities computed on the learned representations. Across all four datasets, the dominant separation appears in the directional term: the factorized model rapidly reaches much lower values, whereas the standard model remains substantially worse under the same angular criterion. By contrast, the radial term remains comparatively close.

This pattern is consistent with the structure of the two objectives. In the standard likelihood, direction and magnitude are entangled, so reducing $\|\boldsymbol{\mu}\|$ can lower the loss without necessarily producing a comparable gain in angular alignment. As this behaviour dominates optimization, posterior variance contracts and the KL rises, diverting capacity toward maintaining a concentrated posterior rather than learning well-structured representations. The factorized formulation removes this shortcut by construction. Its directional term depends only on the position on the unit sphere, so absolute norm reduction cannot improve it, while its radial term depends only on the *difference* in norms between views rather than on absolute magnitude. The resulting optimization pressure is therefore aligned with the intended geometry of the problem, as directional agreement must be improved through better angular structure, and radial consistency through tighter norm matching. Taken together, Table 11, Figure 7, and Figure 8 indicate that the pathology is systematic rather than incidental, while suggesting that the factorized likelihood removes the underlying imbalance rather than merely reducing its visible consequences.

### C.2 Monte Carlo sample ablation

The NLL loss in Eq. (4) is estimated with $K$ reparameterized samples per training step. We sweep $K \in \{0, 1, 2, 4, 16, 64\}$ on CIFAR–10, CIFAR–100, and STL–10 under the standard 200-epoch configuration

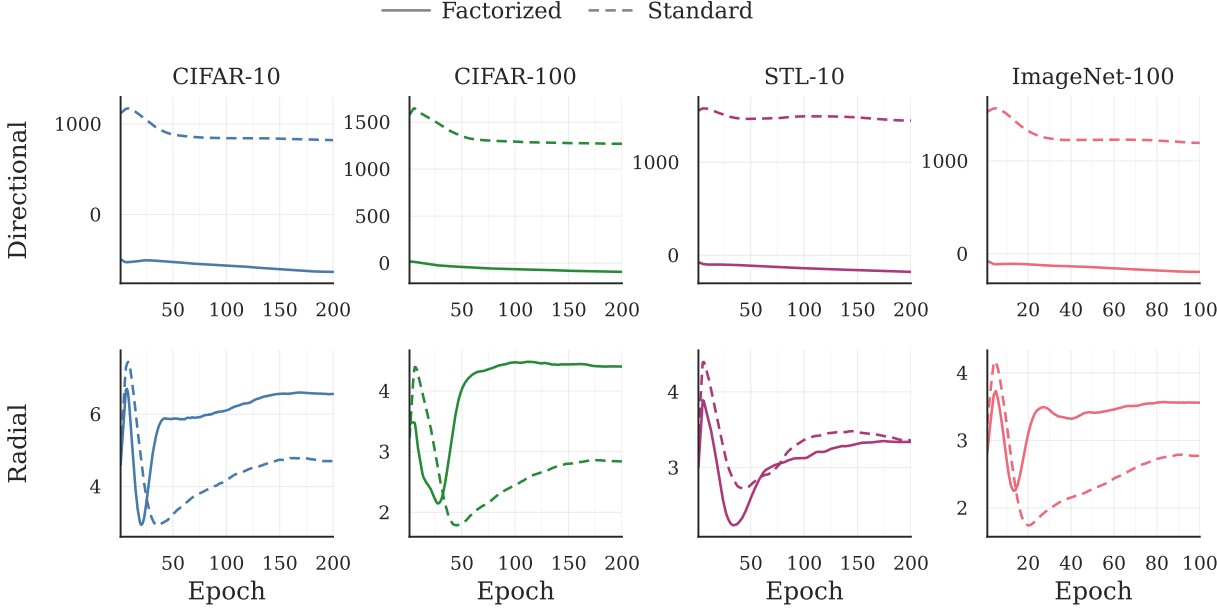

Figure 8: Directional and radial diagnostics obtained by evaluating both likelihood variants under the same factorized decomposition. Columns correspond to datasets; the top row shows the directional diagnostic and the bottom row the radial diagnostic. The standard formulation remains consistently worse under the directional diagnostic, while differences in the radial diagnostic are comparatively modest.

Table 12: Effect of Monte Carlo samples $K$ on three datasets (200 epochs, three seeds). $K=0$ substitutes the posterior mean $\boldsymbol{\mu}$ for the reparameterized sample. All $K \geq 1$ configurations achieve indistinguishable accuracy and posterior variance, while $K=0$ reverts toward the prior.

| | CIFAR–10 | | CIFAR–100 | | STL–10 | |
|---|---|---|---|---|---|---|
| $K$ | $k$–NN | $\bar{\sigma}^2$ | $k$–NN | $\bar{\sigma}^2$ | $k$–NN | $\bar{\sigma}^2$ |
| 0 ($\boldsymbol{\mu}$) | 34.3±0.6 | 0.876 | 10.0±0.2 | 0.865 | 10.6±1.4 | 0.925 |
| 1 | 87.3±0.2 | 0.647 | 56.3±0.2 | 0.653 | 80.9±0.1 | 0.658 |
| 2 | 87.3±0.2 | 0.647 | 56.3±0.1 | 0.655 | 80.9±0.1 | 0.657 |
| 4 | 87.2±0.2 | 0.646 | 55.9±0.5 | 0.652 | 80.8±0.2 | 0.658 |
| 16 | 87.3±0.1 | 0.648 | 55.6±1.1 | 0.650 | 81.2±0.4 | 0.657 |
| 64 | 87.4±0.1 | 0.646 | 56.4±0.2 | 0.655 | 81.0±0.2 | 0.657 |

($\nu=1.0$, $\beta=1.0$, three seeds). The setting $K=0$ replaces the reparameterized sample with the posterior mean $\boldsymbol{\mu}$, eliminating stochasticity from the forward pass entirely. Table 12 reports $k$–NN accuracy and the dimension-averaged posterior variance $\bar{\sigma}^2 = \frac{1}{D} \sum_{d=1}^{D} \sigma_d^2$.

Increasing $K$ from 1 to 64 provides no measurable benefit (Table 12), confirming that the ELBO gradients are well-approximated by a single reparameterized sample. Accordingly, all experiments in the main text use $K=1$.

The $K=0$ setting exhibits signs of collapse, as it reverts toward the prior and yields near-chance accuracy on CIFAR–100 and STL–10, with elevated posterior variance ($\bar{\sigma}^2 \approx 0.87$–$0.93$) approaching the unit prior. On CIFAR–10, partial learning survives (34.3%), suggesting that the deterministic posterior mean carries some alignment signal on simpler tasks but is insufficient for learning a specialized posterior. This pattern is expected, since without reparameterized samples the likelihood gradients do not flow through $\boldsymbol{\sigma}^2$, leaving the KL term as the sole influence on the variance, and since the KL drives $\boldsymbol{\sigma}^2$ toward the prior rather than away from it, the posterior cannot specialize.

### C.3 Assessment of isotropic framework construction

To test whether feature-wise variance is necessary, we replace the diagonal posterior:

$$q(\mathbf{s} \mid \mathbf{z}) = \mathcal{N}\big(\boldsymbol{\mu}, \mathrm{diag}(\boldsymbol{\sigma}^2)\big) \tag{49}$$

with an isotropic scalar-variance variant:

$$q(\mathbf{s} \mid \mathbf{z}) = \mathcal{N}(\boldsymbol{\mu}, \alpha \mathbf{I}), \tag{50}$$

where $\alpha > 0$ is a single learned variance per sample. The same scalar variance replaces the diagonal scale matrix in the directional likelihood, so this variant removes both feature-wise anisotropy and any possibility of allocating uncertainty unevenly across dimensions. All other settings are kept identical to the main experiments (200 epochs, ResNet–18, $\beta=1.0$).

Table 13 shows that this isotropic variant fails across all tested values of $\nu$ on CIFAR–10, CIFAR–100, and STL–10. These results are from single diagnostic runs, but the failure is uniform across all tested values of $\nu$. The posterior mean $\boldsymbol{\mu}$ remains at chance-level $k$–NN accuracy (10.0% on CIFAR–10 and STL–10, 1.0% on CIFAR–100) for every $\nu$, so the inference network does not produce class-informative latent means.

The collapse is visible directly in the learned scalar variance. For all tested $\nu$ and all three datasets, the batch standard deviation of $\alpha$ converges to zero to numerical precision, with coefficient of variation $\mathrm{CoV}(\alpha) = 0$. The isotropic posterior therefore does not carry instance-level uncertainty, as it degenerates to a single global scale shared across the batch.

Under the same training protocol, anisotropic VJE learns discriminative encoder features and an informative posterior mean, whereas the isotropic variant converges to a constant-variance posterior whose mean is

Table 13: Isotropic VJE on CIFAR–10, CIFAR–100, and STL–10 using a scalar posterior variance $\alpha$ with covariance $\alpha\mathbf{I}$ (200 epochs, ResNet–18). In the isotropic case, $\overline{\sigma^2} = \alpha$. The $k$–NN columns report accuracy (%) on the encoder output $z$ and posterior mean $\boldsymbol{\mu}$. Values are from single diagnostic runs. For all isotropic configurations, the batch standard deviation and coefficient of variation of $\alpha$ are zero to numerical precision.

| Dataset | $\nu$ | $k$–NN (%) $z$ | $\boldsymbol{\mu}$ | $\overline{\sigma^2}$ |
|---------|-------|------|------|------|
| CIFAR–10 | 1 | 20.5 | 10.0 | 0.157 |
|  | 3 | 26.5 | 10.0 | 0.239 |
|  | 7 | 24.6 | 10.0 | 0.155 |
|  | 20 | 26.8 | 10.0 | 0.077 |
|  | $\infty$ | 23.5 | 10.0 | 0.022 |
| CIFAR–100 | 1 | 7.3 | 1.0 | 0.443 |
|  | 3 | 6.5 | 1.0 | 0.235 |
|  | 7 | 6.7 | 1.0 | 0.133 |
|  | 20 | 7.3 | 1.0 | 0.071 |
|  | $\infty$ | 5.6 | 1.0 | 0.022 |
| STL–10 | 1 | 11.1 | 10.0 | 0.055 |
|  | 3 | 9.2 | 10.0 | 0.035 |
|  | 7 | 10.5 | 10.0 | 0.027 |
|  | 20 | 9.5 | 10.0 | 0.022 |
|  | $\infty$ | 10.1 | 10.0 | 0.018 |
| *Anisotropic reference ($\nu$=1.0)* | | | | |
| CIFAR–10 | 1 | 87.3 | 87.0 | 0.647 |
| CIFAR–100 | 1 | 56.3 | 55.2 | 0.655 |
| STL–10 | 1 | 80.9 | 80.3 | 0.657 |

uninformative. These results indicate that the VJE formulation requires feature-wise variance in order to represent useful posterior structure.

## C.4 Compute profile

We assess the computational and memory requirements of VJE relative to non-contrastive baselines at ImageNet scale. Table 14 reports parameter counts, per-image floating-point operations (GFLOPs), peak memory, and relative wall-clock throughput for a ResNet–50 encoder at ImageNet resolution (224×224). Parameter counts and GFLOP estimates at a given input resolution are properties of the model and counting convention, while measured peak memory and throughput additionally depend on implementation details and hardware/runtime conditions. Relative throughput is measured on a single NVIDIA GH200 GPU at batch size 256 (averaged over three profiling runs) and normalized by SimSiam.

VJE's inference network adds 3.4M parameters beyond the encoder, compared to 14.7M for SimSiam's projector–predictor stack, 11.6M for BYOL's projector–predictor, and 151.0M for VICReg's expander (Table 14). The per-step cost is governed primarily by the number of encoder passes: VJE and SimSiam each perform two and match at 49.1 vs. 49.2 GFLOPs/img (0.98× relative throughput), while VJE with EMA and BYOL each perform four and match at 65.4 vs. 65.6 GFLOPs/img (0.74× vs. 0.75×). Within each pair, forward-pass FLOPs are also effectively identical (16.4G for both VJE and SimSiam; 32.7G vs. 32.8G for VJE with EMA and BYOL), confirming that the head contribution is negligible relative to the encoder. The computational overhead of VJE's variational loss is therefore mostly absorbed by its smaller head, yielding a per-step cost indistinguishable from that of the corresponding deterministic baselines.

Table 14: Computational profile with a ResNet–50 encoder at ImageNet resolution. *Encoder passes* counts forward passes through the encoder per training step; EMA methods additionally perform no-gradient target passes, which are cheaper than gradient-tracked source passes (the step cost of four-pass methods is therefore less than double that of two-pass methods). *Head params* excludes the shared encoder (23.5M for all methods). *Fwd GFLOPs/img* and *Step GFLOPs/img* are per-image costs of one forward pass and one full training step (forward + backward) respectively, estimated via `torch.utils.flop_counter`. *Relative throughput* is wall-clock speed normalized to SimSiam = 1.00. *Peak memory* is measured in float32; entries marked [†] are linearly extrapolated from measurements at the method's reference batch size.

|  | VJE | VJE (EMA) | SimSiam | BYOL | VICReg |
|---|---|---|---|---|---|
| *Parameters* | | | | | |
| Head params (M) | 3.4 | 3.4 | 14.7 | 11.6 | 151.0 |
| Trainable params (M) | 26.9 | 26.9 | 38.2 | 35.1 | 174.5 |
| Total params (M) | 26.9 | 50.4 | 38.2 | 68.0 | 174.5 |
| *Compute* | | | | | |
| Encoder passes / step | 2 | 4 | 2 | 4 | 2 |
| Fwd GFLOPs/img | 16.4 | 32.7 | 16.4 | 32.8 | 17.2 |
| Step GFLOPs/img | 49.1 | 65.4 | 49.2 | 65.6 | 51.7 |
| Relative throughput | 0.98× | 0.74× | 1.00× | 0.75× | 0.94× |
| *Memory* | | | | | |
| Peak @256 (GB) | 41.4 | 43.7 | 41.5 | 43.8 | 43.0 |
| Peak @ref. BS (GB) | 41.4 | 43.7 | 41.5 | 696.0[†] | 330.5[†] |
| Reference batch size | 256 | 256 | 256 | 4096 | 2048 |

At a uniform batch size of 256, peak memory is comparable across all methods (41–44 GB). However, BYOL and VICReg require reference batch sizes of 4096 and 2048 respectively for their published performance; the corresponding peak-memory entries in Table 14 are linearly extrapolated to 696 GB and 331 GB and would in practice require multi-device training. VJE (including the EMA variant) and SimSiam reach their reported configurations at batch size 256 without such constraints.

