# OpenReview forum: "Joint Embedding Variational Bayes"
_TMLR — Accepted by TMLR_

### Review · Reviewer_BcRg · 2026-02-07

**Summary Of Contributions:**

This paper introduces Variational Joint Embedding, a framework that combines joint embedding architectures with variational inference for self-supervised learning. The key contributions are:

1.  VJE maximizes a symmetric conditional ELBO on encoder embeddings, providing a principled alternative to energy-based predictive objectives
2.  The model explicitly decouples directional and radial components using heavy-tailed distributions to prevent norm-induced instabilities
3.  Feature-wise variances are tied between the variational posterior and directional likelihood, enabling anisotropic uncertainty without projection heads
4.  Competitive performance on standard SSL benchmarks (ImageNet-1K, CIFAR-10/100, STL-10) and strong anomaly detection results (90.3% AUROC on CIFAR-10)

Key Strengths:
- Well-motivated theoretical framework with clear distinction from energy-based objectives (Appendix B is particularly strong)
- Comprehensive experimental evaluation including both discriminative and probabilistic assessments
- Thorough ablation studies demonstrating the necessity of each component
- Strong anomaly detection performance validates the probabilistic semantics
- Clear mathematical exposition and detailed derivations

Key Weaknesses:
- Performance gap on ImageNet-1K (65.6% vs 68.6% for VICReg) suggests trade-offs between probabilistic modeling and discriminative power
- Limited scalability analysis (only ResNet architectures tested, no ViT or larger models)
- No computational cost analysis compared to deterministic baselines
- Scope limited to vision tasks; generalization to other modalities unexplored
- Missing comparisons to some recent probabilistic SSL methods

**Audience:**

Yes

**Audience Explanation:**

The work bridges deterministic and probabilistic approaches, offering a principled alternative to heuristic non-contrastive methods

Demonstrates that meaningful uncertainty can be learned without pixel-level reconstruction, opening new directions for efficient probabilistic representation learning

Provides a strong SSL-based anomaly detection method (90.3% AUROC) that outperforms comparable self-supervised baselines

The objective-level distinction between energy-based and likelihood-based formulations (Appendix B) provides valuable conceptual clarity

The framework offers uncertainty estimates "for free" during SSL pretraining, which is valuable for downstream applications requiring calibrated predictions

**Broader Impact Concerns:**

The paper does not currently include a Broader Impact Statement, but the concerns are minimal. The work is primarily methodological with limited direct ethical implications.

**Claims And Evidence:**

Yes

**Claims Explanation:**

The framework is rigorously developed with clear derivations. The polar decomposition and its motivation through isotropic Student-t factorization (Section 4.1) is well-justified.

 The experiments adequately support the main claims:
   - Table 1 shows competitive (though not state-of-the-art) performance on ImageNet-1K
   - Table 2 demonstrates parity with SimSiam and superiority over VICReg on CIFAR-10
   - Table 4 and Figure 4 provide strong evidence for meaningful probabilistic semantics through anomaly detection
   - Ablations (Tables 5-7) convincingly demonstrate the necessity of Student-t likelihoods and each loss component

The heavy-tailed likelihood choice is well-motivated both theoretically (Section 4, Figure 2) and empirically (Table 5 shows Gaussian likelihood failure)

Minor concerns:
- The claim of "comparable performance" on ImageNet-1K is somewhat generous given the 2-3% gap to leading methods
- Some experimental details could be clearer (e.g., how many samples for Monte Carlo in practice?)
- The paper would benefit from reporting confidence intervals for more experiments beyond ImageNet-1K

**Requested Changes:**

Add a table comparing training time, memory usage, and throughput (images/sec) against SimSiam and VICReg baselines. This is essential for understanding the practical trade-offs of the probabilistic formulation.

Reporrt standard deviations for the ImageNet-1K and CIFAR experiments (currently only provided for ImageNet-1K). This is important for assessing statistical significance of comparisons.

Explicitly state in Section 5.1 that single-sample Monte Carlo is used throughout (currently only mentioned in Appendix A). This affects reproducibility and interpretation.

Missing related work: The paper should cite and discuss:
   - [1] on spectral-normalized neural Gaussian processes for uncertainty quantification

   - [2] on variational contrastive learning with beta-divergence

   - Recent work on variational supervised contrastive learning [3]

Add experiments with larger backbones (ResNet-101, ViT-S/B) on ImageNet-1K to demonstrate scalability. This would address concerns about the method's applicability to modern architectures.

Add t-SNE or UMAP visualizations of embeddings colored by uncertainty (variance magnitude) to provide intuition about what the model learns. Include comparisons of in-distribution vs. out-of-distribution samples.

Include reliability diagrams or ECE for downstream classification to validate that the learned uncertainties are well-calibrated, not just useful for anomaly detection.

Investigate whether using K>1 samples during training improves performance, and provide guidance on this trade-off.

Discuss which CIFAR-10 classes perform poorly in anomaly detection (Table 3 shows variation from 0.828 to 0.976) and provide hypotheses for why certain classes are harder.

While beta and nu are swept extensively for anomaly detection, provide guidance on how to select these for discriminative tasks. Is (beta=1, nu=3) generally recommended?

Perhaps provide sample complexity or generalization bounds relating the ELBO optimization to downstream task performance.

[1] Liu J, Lin Z, Padhy S, Tran D, Bedrax Weiss T, Lakshminarayanan B. Simple and principled uncertainty estimation with deterministic deep learning via distance awareness. Advances in neural information processing systems. 2020;33:7498-512.

[2] Yavuz MC, Yanikoglu B. Variational self-supervised contrastive learning using beta divergence. arXiv preprint arXiv:2312.00824. 2023 Sep 5.

[3] Wang Z, Fan J, Nguyen T, Ji H, Liu G. Variational Supervised Contrastive Learning. arXiv preprint arXiv:2506.07413. 2025 Jun 9.

---

> ### Author Response · Authors · 2026-03-30
>
> Thank you for your review and the constructive suggestions. The revision incorporates many of them directly.
>
> We added the practical-cost comparison you requested. Appendix C.4 now reports head parameters, total/trainable parameters, forward and step GFLOPs per image, peak memory, and relative throughput against SimSiam, BYOL, and VICReg. This makes the computational trade-off of the probabilistic formulation explicit.
>
> We also added variability reporting beyond the original ImageNet table. The revised CIFAR/STL representation-learning results, the OOD tables, and the ablations now report means and standard deviations over repeated runs. In addition, Section 5.1 now states directly in the main text that K=1 reparameterized sampling is used throughout, and Appendix C.2 provides the requested K-sweep. That ablation shows no measurable benefit from K>1 in our setting, so the revision now provides clear guidance that K=1 is sufficient.
>
> The related-work section was expanded to cover the uncertainty-aware and variational representation-learning papers you flagged. We also added a direct empirical comparison against VI-SimSiam in the revised OOD section.
>
> On visualization and semantic intuition, the revision now includes a dedicated "Geometry of learned uncertainty" section. This adds t-SNE visualizations colored by posterior-derived quantities and a quantitative analysis relating uncertainty structure to class-center margin and within-class radius. We also added OOD ROC curves and score-distribution plots, so the revised paper now gives much more intuition about what the posterior is encoding.
>
> We did not add ViT or larger-backbone experiments in this revision. We agree scalability is important, but for VJE this is not simply a matter of swapping the backbone, since the current observation model and inference network are defined on a single global embedding, whereas a meaningful ViT extension would require a token-level or hierarchical likelihood design. We therefore discuss this explicitly as future work rather than presenting an incomplete backbone-only comparison, or a partial scalability study.
>
> We also did not add downstream reliability diagrams or ECE. Our uncertainty in this paper is intrinsic to representation space, and so turning it into calibrated predictive uncertainty for a supervised classifier requires an additional task-space mapping and a different evaluation protocol. We now note this limitation and future direction explicitly in the conclusion.
>
> Finally, Appendix C.1 now compares the factorized likelihood against the unfactorized "standard" formulation and shows the pathology empirically, especially at larger scale. This gives direct  support for the modelling choice that was only theoretically motivated in the submitted paper.

---

### Review · Reviewer_79XR · 2026-03-06

**Summary Of Contributions:**

The authors introduce variational joint embedding (VJE), which implements a latent-variable formulation of self-supervised learning without the need for reconstruction (decoding). Moreover, instead of optimizing for differences between (paired) embeddings, they define a likelihood on target embeddings. Their model is trained by optimizing a symmetric conditional ELBO on paired embeddings, in which the target branch is detached to implement conditioning on the likelihood. In their experiments, VJE performs comparably to other non-contrastive approaches on image datasets.

**Additional Comments:**

NA

**Audience:**

Yes

**Audience Explanation:**

The main topics of the paper, namely self-supervised learning and variational inference are clearly attractive to the readership of TMLR. Moreover, the proposed approach is an interesting take on variational inference for latent variable models in the context of self-supervised learning.

**Broader Impact Concerns:**

No concerns.

**Claims And Evidence:**

Yes

**Claims Explanation:**

The methodological claims made by the authors are supported, however, the results do not seem to support that the proposed model is better than existing alternatives or that it provides an alternative value or trade-off relative to existing approaches.

**Requested Changes:**

It is unclear why the analysis of the likelihood distribution in Section 4 is done with a fixed \lambda instead of using the same approach as in (7), where the likelihood is calculated as p(s|z) rather than p(z|s) as in (11). Moreover, arguably, the latter is a likelihood, while the former is not. Are the results in Figure 2 calculated the formulation in (12) or (7)?

The results in Section 5.2 require additional clarification. Specifically, was the ResNet-50 trained from random weights or initialized with a pre-trained model? the question is because ResNet is typically trained on ImageNet data. What backbone was used for the baselines? without clarity on those is difficult to assess how comparable results are because it is not clear whether performance is due to the backbone, the training methodology or both. For the results in Table 2, it is not clear why error bars are not provided. From Figure 3, it looks like both SimSiam and VJE can benefit from additional runtime, is there a justification from stopping the training at 800 epochs?

The results in Section 5.3 are welcome, however, these also highlight the need for tuning the degrees-of-freedom parameter nu, which is especially difficult in anomaly detection applications (if done properly, i.e., in an unsupervised fashion). Note that there is a difference in AUC between the optimal (nu=0.5) and the parameters used in Section 5.2 (nu=3.0) of 0.024. This could also be used as a justification for the model to be insensitive to the value of its hyperparameters, however, it will make the model underperform in Table 4. Also, assuming that results in Figure 4 are averages over 10 classes, including the error bars will be useful and having a separate row in Table 3 showing the best AUC for each class combination will also be useful to assess how generally "optimal" is the (1.0,0.5) parameter combination.

For the results in Section 4, are the results for Gaussian using the same factorized likelihood and formulation as the proposed method or rather using the formulation in (12)? if the latter, why is it the case?

It is not clear from the results in Table 6 that the rad term in the loss is significantly impacting performance after accounting for error bars (which are not presented). Results will be more convincing with error bars justifying the significance of the gains.

The purpose of the results in Table 7 are not entirely clear.

In summary, the proposed method is an intriguing application for variational inference for latent variable models in the context of self-supervised learning and with a factorized likelihood using a heavy tailed distribution. From a methodology perspective, it is not entire clear why the factorization of the likelihood is necessary, which also seems supported by the results in Table 6. The experiments are for the most part underwhelming because the proposed method only performs comparably to existing approaches and it does not seem to claim any advantages over them either in terms of computational complexity, ease of training, interpretation, use cases, etc.

Minor:
- Point to the definitions of undefined terms in Figure 1: \hat{z}, \ell_dir, \ell_rad.
- \hat{z} is used before being defined in (6).
- There is some unnecessary re-definition of quantities, for instance, q() in the text and (32); s in (3), the text and again (33); the KL term in (9) and (34); the loss in (6)(10) and (38). This repetition also makes Section 4.4 redundant from the perspective of the formulation, which has been introduced and described elsewhere.
- Table 3 is not referenced in the text.

---

> ### Author Response · Authors · 2026-03-30
>
> Thank you for the detailed and careful reading. Several of the ambiguities you noted came from the submitted paper mixing together motivational likelihood analysis, the final training objective, and the semantics of the conditional model. The revision addresses this at both the theoretical and experimental levels.
>
> First, on the modelling roles and the likelihood discussion in Section 4, the revised manuscript now states consistently that q(s|z) is the amortized variational posterior and that the generative scoring model is the conditional model on the reparameterized observation y=(\hat z, ||z||). The scalar Gaussian/Student-t discussion is now framed strictly as motivation for the likelihood family, and Figure 2 is labelled accordingly as a one-dimensional illustration of the likelihood behaviour. We also clarify that all references to the "Gaussian" case mean the \nu -> \infty limit within the same Student-t family. Most importantly, the directional construction itself has been replaced by the intrinsic tangent-space formulation described in Section 4.2, so the revised text no longer relies on the problematic extrinsic normalization claim from the submission.
>
> Second, on the experimental setup, Section 5.1 now states explicitly that all models are trained from random initialization. It also clarifies which parts of the setup are shared across methods and which remain method-specific for reproduced baselines. We added standard deviations to the revised CIFAR/STL and OOD tables, and the revised multi-dataset comparison now reports means and std over three runs. The revised ImageNet experiment also changes materially: the final manuscript uses an EMA target encoder and reports the corresponding result under the revised formulation.
>
> Third, on sensitivity to \nu and experimental guidance, the revision separates representation-learning sensitivity from uncertainty-scoring sensitivity. For discriminative representation learning, performance is fairly stable across a broad range of finite \nu values; for OOD scoring, smaller \nu values are consistently preferred and large \nu eventually collapse toward the Gaussian limit. Based on these results, the revised paper adopts \nu = 1.0 as the default and presents explicit sweeps in both the main paper and appendices.
>
> Fourth, on the old anomaly-detection section, we ultimately decided not to keep the submitted one-class CIFAR-10 experiment in the revised manuscript. Rather than partially updating that section, we replaced it with a broader OpenOOD evaluation under the revised formulation, together with score comparisons, ROC curves, density plots, and a reproduced comparison to VI-SimSiam.
>
> Fifth, on the role of the radial term and the ablations, the revised loss ablation now makes the interpretation more explicit. The results show that L_dir + L_KL is nearly indistinguishable from the full objective for representation accuracy, so the manuscript now states clearly that the radial term acts mainly as a geometric anchor rather than the primary source of discriminative gain. This was an important clarification suggested by your review.
>
> We also cleaned up a number of minor issues you noted: Figure 1 now defines the observation more clearly, K=1 is stated in the main text, redundant re-definitions were reduced, and cross-references were tightened so that the notation is introduced in one place and reused more consistently.

---

### Review · Reviewer_WmXF · 2026-03-10

**Summary Of Contributions:**

The paper proposes Variational Joint Embedding (VJE), a non-contrastive self-supervised method that replaces the deterministic matching loss between two augmented views with a symmetric conditional ELBO defined directly in embedding space. The paper argues that a plain Gaussian or Euclidean discrepancy entangles angular disagreement and norm mismatch, which can produce unstable gradients. To address this, it decomposes the representation into direction and magnitude, uses a heavy-tailed Student–t distrtibution for robustness, shares the same diagonal variance vector between the posterior and the directional likelihood, and fixes the radial scale. They show the Gaussian limit performs very poorly, and that removing the KL term collapses the posterior variance, while removing the directional term makes the posterior match an isotropic prior. That supports the paper’s claim that the heavy-tailed directional term plus KL are doing most of the important work, while the radial term mainly acts as a geometric anchor.

**Audience:**

Yes

**Audience Explanation:**

The problem studied in the paper is in general important in ML community.

**Claims And Evidence:**

No

**Claims Explanation:**

The paper supports some of its claims clearly and fairly well, but its strongest theoretical claims are not fully justified, and some of the empirical evidence is suggestive rather than fully convincing. The clearest supported claim is that VJE can learn reasonable non-contrastive representations.The claim about probabilistic usefulness for anomaly detection is also supported to a degree. The one-class CIFAR-10 experiment is aligned with the paper’s modeling goal, and the paper shows that the joint likelihood score beats its own entropy and variance alternatives and slightly exceeds the generic baselines it compares against.

However, the paper becomes less convincing in the claim that VJE provides a 'normalized probabilistic formulation' in representation space. The isotropic polar decomposition is cleanly motivated, but when the paper moves to the actual directional model it explicitly switches to an extrinsic construction in ambient space and says the sphere Jacobian is 'absorbed into the normalizing constant.' I do not think the paper proves that this remains a properly normalized directional density on the unit sphere once anisotropic diagonal whitening is introduced. That weakens one of the central theoretical claims.

**Requested Changes:**

How sensitive are results to replacing the diagonal $\sigma^2$ with a scalar or isotropic scale? That would directly test whether feature-wise uncertainty is genuinely helpful.

Can you provide a fair comparison against VI-SimSiam or VSSL, even on a smaller benchmark?

Can you report compute and parameter overhead relative to SimSiam/VICReg, since the paper argues it avoids auxiliary projectors but still adds an inference network and sampling path?

Lastly, there are some related work that should be discussed:
[1] Ziwen Wang, et al. "Variational Supervised Contrastive Learning"
[2] Minoh Jeong, et al. "Probabilistic Variational Contrastive Learning"

---

> ### Author Response · Authors · 2026-03-30
>
> Thank you for your review. You identified a genuine issue with the submitted formulation, as we incorrectly conflated parameter-independent constants, with the normalizing measure on the sphere, which does indeed depend on the learned variance after anisotropic whitening. The submitted extrinsic construction was therefore not a proper normalized likelihood.
>
> To address this, we made a substantial change and replaced the extrinsic formulation entirely with an intrinsic construction on \(S^{D-1}\), developed in the rewritten Section~4.2. The target direction is mapped by the geodesic log-map, the ambient diagonal covariance is restricted to the tangent space through a Schur-complement construction, and the exponential-map Jacobian accounts for curvature. The paper now distinguishes clearly between (i) a normalized spherical reference construction and (ii) the directional subdensity actually optimized in training, with the ELBO stated for the associated subprobability model. All experiments were rerun under this revised formulation.
>
> We also added the isotropic-vs-diagonal comparison you requested. Appendix C.3 replaces the diagonal posterior with a scalar-variance isotropic variant and shows uniform failure across CIFAR-10, CIFAR-100, and STL-10: the posterior mean stays at chance level and the learned scalar variance collapses to a constant. This supports the claim that feature-wise uncertainty is not cosmetic but necessary for the framework to learn useful posterior structure.
>
> On empirical comparisons, the revised manuscript now includes a direct reproduced comparison against VI-SimSiam in the OOD section, both on the OpenOOD CIFAR-10 benchmark and on the shared SVHN/CIFAR-100 benchmark. We did not include VSSL in the final comparison because we were not able to obtain a stable, reliable reproduction under its published setup, and we preferred not to report an untrustworthy comparison.
>
> We also added the practical-cost comparison you asked for. Appendix C.4 reports parameter counts, per-image GFLOPs, peak memory, and relative throughput against SimSiam, BYOL, and VICReg. This shows that VJE does avoid large projector stacks, but still incurs the expected inference-network and sampling overhead; the revised manuscript now makes that trade-off explicit.
>
> Lastly, we expanded the related-work discussion to include the missing variational and uncertainty-aware representation-learning papers you highlighted.

---

### Author Response · Authors · 2026-03-30
**Note on Revisions**

We thank the reviewers for the careful and constructive feedback. We have made a substantial revision in response. The most important change is theoretical: the submitted extrinsic directional likelihood has been replaced with an intrinsic tangent-space construction on the sphere. The revised paper now presents a normalized spherical reference model, then derives the directional subdensity actually used in training, and states the ELBO explicitly for the associated subprobability model. This resolves the normalization issue in the original submission and makes the probabilistic semantics precise.

We also reran the experiments from scratch under the revised formulation and expanded the empirical section considerably. In the main paper, we now: (i) clarify that all models are trained from random initialization; (ii) state directly that K=1 Monte Carlo sampling is used throughout; (iii) add standard deviations across runs beyond the ImageNet result; (iv) add an EMA target-encoder variant and report improved ImageNet-1K performance; (v) replace the previous one-class anomaly section with a broader OpenOOD evaluation; (vi) add a geometry-of-uncertainty analysis with t-SNE visualizations and quantitative margin/radius correlations; and (vii) add direct comparisons against VI-SimSiam.

The appendices now include additional diagnostics and implementation details: a standard-vs-factorized likelihood comparison showing collapse of the unfactorized formulation, a Monte Carlo sample ablation, an isotropic scalar-variance ablation, and a compute profile comparing parameters, FLOPs, memory, and throughput against standard non-contrastive baselines. We also expanded the related-work discussion to cover the missing uncertainty-aware and variational contrastive literature highlighted by the reviewers.

---

### Decision · Action_Editor_5hgw · 2026-04-15

**Recommendation:** Accept as is

**Audience:**

Yes

**Audience Explanation:**

Generative modeling is a central topic in machine learning. This paper introduces an extended framework Variational Joint Embedding (VJE) that might be a new advantageous way to model data.

**Claims And Evidence:**

Yes

**Claims Explanation:**

All reviewers agree that this paper is sound and convincing.